# Age-associated decline of Coenzyme A leads to intestinal stem cells dysfunction via disturbing iron homeostasis

Zhiming Liu[1◉], Gang Du[2,3◉], Yi Chen[4*], Haiyang Chen[1*]

**1** West China Centre of Excellence for Pancreatitis, and Laboratory of Metabolism and Aging, Frontiers Science Center for Disease-related Molecular Network, State Key Laboratory of Respiratory Health and Multimorbidity and National Clinical Research Center for Geriatrics, West China Hospital, Sichuan University, Chengdu, China, **2** Department of Biological Chemistry and Molecular Pharmacology, Harvard Medical School, Boston, Massachusetts, United States of America, **3** Program in Cellular and Molecular Medicine, Boston Children's Hospital, Boston, Massachusetts, United States of America, **4** Division of Gastrointestinal Surgery, Department of General Surgery, West China Hospital, Sichuan University, Chengdu, Sichuan, China

◉ These authors contributed equally to this work.
* chenhy82@scu.edu.cn (HC); toddy@scu.edu.cn (YC)

## Abstract

The decline in adult stem cell performance is closely linked to tissue malfunction and the rising incidence of age-related diseases. To investigate the molecular basis of these impairments, our screening strategy identified reduced activity in the pantothenate/coenzyme A (CoA) pathway within aged ISCs. Furthermore, exogenous CoA supplementation restructured ISC metabolic pathways, reversing age-induced hyperproliferation and intestinal dysfunction, and thus extending *Drosophila* lifespan by curbing excessive iron accumulation in ISCs. These findings uncover a new mechanism of stem cell aging and propose that pantothenate and CoA could be potential therapeutic targets for treating age-related diseases and enhancing healthy aging in humans.

### Author summary

In our study, we explored the connection between the aging of adult stem cells and the rise of age-related diseases, uncovering a significant metabolic link. As stem cells age, a key metabolic pathway involving pantothenate and CoA becomes less active, leading to an accumulation of excess iron, which disrupts cellular function and contributes to the aging process. To counteract this decline, we added pantothenate and CoA to the aging stem cells, restoring their normal function, reducing iron buildup, and ultimately improving the health and lifespan of old fruit flies. This simple intervention not only highlights the potential of pantothenate and CoA as powerful tools to combat the effects of aging but also

**Data availability statement:** The RNA-seq data for CoA-treated T cells and untreated controls (GSM5668528–GSM5668533) were obtained from the GEO repository (GSE187456) and are available for further analysis. The complete dataset of single-cell RNA-seq data, including gene expression profiles and metadata, is accessible via the AFCA portal https://hong-jielilab.org/afca. AFCA Contact Email: hongjie.li@bcm.edu. This platform allows users to explore gene-specific changes during aging and provides interactive tools for data analysis. Researchers can freely access aging trajectories and download necessary analytical assets through the AFCA interface.

**Funding:** This work was supported by the National Key R&D Program of China (2020YFA0803602 to H.C.), the National Natural Science Foundation of China (32470879, 92157109 to H.C.), and the 1·3·5project for disciplines of excellence, West China Hospital, Sichuan University (ZYYC20024 to H.C.). 1·3·5 projects for Artificial Intelligence, West China Hospital, Sichuan University (ZYAI24024 to Y.C.), Sichuan Science and Technology Program, Sichuan Provincial Natural Science Foundation (Grant No. 2025ZNSFSC0720 to H.C.), Natural Science Foundation of Sichuan Province (Grant No. 2023NSFSC0664, China to Y.C.), Noncommunicable Chronic Diseases-National Science and Technology Major Project (2023ZD0506800 to H.C.). The funders had no role in study design, data collection and analysis, decision to publish, or preparation of the manuscript.

**Competing interests:** The authors have declared that no competing interests exist.

suggests broader implications for treating age-related diseases and promoting healthy aging in humans. Our findings provide a new perspective on the biology of aging and open up exciting new avenues for therapeutic interventions that could improve the quality of life for older individuals.

## Introduction

The incidence of tissue malfunction, illness, and various cancers increases exponentially with age. Studies across various experimental models show that the reduced functionality of endogenous stem cells may contribute to the deterioration and impaired functioning of aged tissues. In recent years, signaling networks, regulatory proteins, and epigenetic mechanisms, particularly those involving chromatin remodeling and proteostasis, are likely pivotal in mediating the changes associated with stem cell senescence and revitalization [1–3]. Despite remarkable progress, the reasons behind the decline in aging-associated stem cell function in tissue maintenance remain poorly understood. Recently, modulating stem cell metabolic pathways to manage cellular energy dynamics has emerged as a potential strategy for enhancing stem cell functionality [4]. However, its role in stem cell aging is still largely unclear.

Metabolic changes consistently impact the intestinal epithelium homeostasis in *Drosophila* and mice, closely associated with cellular aging [5]. Notably, NAD+ replenishment has been observed to delay stem cell senescence, possibly mediated by the enzyme Sirt1 and related to the regulation of mitophagy, as recent studies suggest [6,7]. Additionally, fasting, as a dietary intervention, has been shown to trigger a metabolic shift associated with extended lifespan and improved aging phenotypes [8]. Specifically, this dietary change promotes the use of triglycerides, enhancing the functionality of aged intestinal stem cells (ISCs) and thus contributing to lifespan extension [9]. Collectively, these findings suggest that modulating stem cell metabolic pathways could slow the aging process of these cells and strengthen tissue functionality in aging organisms.

This study was inspired by our previous screening results involving the use of flies that have been fed a diet containing a variety of small molecule compounds. These results showed a decreasing trend in the number of ISCs in aged flies following CoA treatment, as indicated by *esg*-driven luciferase expression [10,11]. Nevertheless, additional research is necessary to validate the findings from our initial screening and explore their implications. The methods through which the majority of cells and organisms obtain adequate CoA concentrations have been comprehended for more than six decades. Pantothenic acid, often referred to as vitamin B5, is taken up by cells and converted into CoA via the action of five enzymes: pantothenate kinase (PanK), phosphopantothenoylcysteine synthase (PPCS), phosphopantothenoylcysteine decarboxylase (PPCDC), phosphopantetheine adenylyltransferase (PPAT), and dephospho-CoA kinase (DPCK) [12,13]. In *Drosophila*, a solitary enzyme responsible for pantothenate kinase activity, known as dPANK/Fbl and categorized under the PANKII family, has been discovered. Disruptions in the CoA biosynthesis route

have serious consequences, including mutations in enzymes associated with this process are linked to inherited human disorders. Such disorders encompass neurodegenerative conditions linked to pantothenate kinase (PKAN) [14], COASY [15], and a lack of PPCS, which is associated with cardiac disease [16]. In colorectal cancer (CRC), primary colon adenocarcinoma exhibits a significantly higher level of pantothenate compared to normal colon tissue [17]. Moreover, the nuclear localization of TMEM120A is critical for its role in regulating chemosensitivity in CRC. Specifically, the nuclear fraction of TMEM120A is essential for maintaining nuclear CoA levels, which subsequently affects nuclear acetyl-CoA and histone acetylation in CRC cells [18]. Additionally, disruptions in the pantothenate and CoA biosynthesis pathways have been noted in CRC patients [19]. This highlights the critical role of the canonical CoA biosynthesis pathway, which is vital for organismal survival. Nevertheless, it remains uncertain if the levels of *dPANK/fbl* expression diminish as ISCs age. Moreover, its role in other organs and tissues aging, especially in aging-associated stem cell dysfunction is not fully understood.

The *Drosophila* midgut's simple cellular architecture, ease of genetic manipulation, and clear stem cell lineage delineation make it an ideal model for studying the age-related decline in stem cell efficacy [20]. This model is crucial for discovering ways to boost the regenerative potential of established stem cells. ISCs in flies are identified by the expression of Delta (Dl), a Notch pathway ligand, and the transcriptional regulator Escargot (*esg*), located in the midgut epithelium's basal layer [21]. These stem cells proliferate to self-renew and differentiate into enteroblast (EB) or enteroendocrine (EE) lineage cells, depending on Notch signaling [22]. The EBs mature into absorptive enterocytes (ECs), while enteroendocrine precursors develop into hormone-secreting EE cells. Typically, the population of ISCs and their precursors is small and balanced in young, healthy midguts, but increases significantly with age [23,24]. This increase is accompanied by a reduced stem cell differentiation capacity, leading to an excess of *esg*- and Dl-expressing cells in the aged midgut [23,24]. Multiple signaling pathways, such as JNK, insulin, mTOR, p38-MAPK, DGF/VEGF, and ROS, are involved in ISC aging and linked to gene expression changes that alter the biological behavior of these cells over time [20,25–27]. These changes can impair gut barrier integrity and disrupt the digestive tract's acid-base balance [28,29]. Notably, interventions that decrease *esg*-expressing cells in the midgut, both genetically and pharmacologically, have been shown to prolong *Drosophila* lifespan [30]. This highlights the fly midgut's value as an experimental model for studying the roles and mechanisms of metabolic pathways in regulating stem cell behavior during aging.

In this study, our goal was to identify new metabolic pathways in ISCs that could counteract the aging-related decline and extend the lifespan of *Drosophila*. Using a screening strategy, we found that the pantothenate/coenzyme A (CoA) pathway is abundant in young ISCs and identified CoA as an inhibitor of age-related ISC over-proliferation. Additionally, we discovered that CoA treatment in flies prevented the age-related functional decline of ISCs and extended *Drosophila* lifespan through iron homeostasis regulation. These findings underscore the pantothenate/CoA pathway as a promising target for enhancing ISC performance in aging and amplifying the effectiveness of anti-aging therapies, potentially promoting human healthy longevity.

## Results

### Orally administrated CoA repressed intestinal epithelial dysplasia and barrier dysfunction of aged *Drosophila*

The intestine of young *Drosophila* contains quiescent intestinal stem cells (ISCs) that can be swiftly triggered in the event of stress or tissue damage and are vital for epithelial regeneration after injury through proper proliferation and differentiation. However, aging leads to defects in regeneration, which is characterized by the reduced ISC differentiation and accumulation of *esg*+ cells in older flies. Metabolism substantially impacts the functionality of ISCs and the equilibrium within the intestine, both aspects being compromised by the aging process. Consequently, the interaction between metabolic processes and aging offers possible targets for intervention to alleviate the age-associated deterioration of the intestinal epithelium.

To demonstrate the impact of CoA on *Drosophila* ISC function in aging flies, we assessed the suppressive effects on age-related accumulation of *esg+* cells utilizing an "*esg*-luciferase" reporter system (Fig 1A and 1B). Supplementing with pantothenate, a precursor to CoA or CoA starting at a moderate age (26 days) demonstrated a significant inhibitory impact on the *esg*-expressing cells accumulation in the midguts of older (40 days) flies (Fig 1C and 1D). In our investigation, we examined three different concentrations of pantothenate supplementation (0.1, 1, and 10 mM), and determined that 1 mM pantothenate, was efficacious in suppressing *esg+* cell accumulation in aged midguts, as evidenced by a substantial reduction in luciferase activity (Fig 1C). Furthermore, our findings revealed that supplementing with 0.5 mM of CoA was similarly effective in inhibiting the *esg+* cells' accumulation within the aged flies' midguts (Fig 1D). Thus, we selected a drug concentration of 1mM for pantothenate and 0.5mM for CoA in the following experiments

To further confirm their anti-aging effects on ISCs, we employed the *esg*-GFP reporter line to evaluate its impact on aged ISCs and their progeny (EBs). We evaluated the number of ISCs identified through Dl staining. In line with earlier results, the quantities of *esg*-GFP +, and Dl+ cells progressively rose as the flies aged in the midguts of *Drosophila*. (Fig 1E–1G and 1J). However, aged flies (26 days) that consumed pantothenate/CoA for 14 days displayed a reduced accumulation of *esg*-GFP positive, and Dl positive cells compared to those that did not receive pantothenate/CoA (Fig 1E–1J). A previous study has shown aging results in decreased production of septate junction proteins (SJPs) within the midgut, thus contributing to the decline in barrier function in older animals [31,32]. Our results showed aged flies with CoA administration displayed a higher Discs large (Dlg, a kind of SJPs) expression compared to control aged flies (Fig 1K and 1L). This result suggested that treatment with CoA could prevent intestinal barrier dysfunction in old flies.

## Exogenous pantothenate/CoA administration prevents ISC functional decline within aged flies

Given that ISC hyperproliferation or disrupted differentiation in aging intestines causes epithelial barrier dysfunction seen in aged flies. It is possible that pantothenate/CoA administration represses epithelium dysplasia and promotes the barrier function of aged flies via inhibiting ISC hyperproliferation or promoting ISC differentiation capability. To test these two possibilities, we first analyzed the ISC proliferation rates indicated by phosphorylated histone3 (pH3 +; an indicator of mitotic activity) immunostaining. Consistent with earlier results, our results showed that the ISC proliferation rate increased significantly in 40-day-old flies (Fig 2A). However, CoA administration at a mid-age significantly repressed ISC proliferation in aged flies (Fig 2A). This finding indicates that CoA administration promotes gut barrier function by repressing ISC hyperproliferation in aged flies.

In the old *Drosophila* intestine, the proportion of pre-EC (*esg*-GFP+ and mex+ cells) and pre-EE (*esg*-GFP+ and pros+ cells) increased significantly, which indicates a block in ISC differentiation. We thus employed a *mex-GAL4, esg-GFP/UAS-mCherry.NLS* line to detect the ratio of pre-EC and found that CoA administration markedly decreased the ratio of pre-EC of old flies (Fig 2B–2E). Moreover, CoA administration also leads to a decreased ratio of pre-EE of old flies (Fig 2F–2I). To further demonstrate that CoA administration promotes ISC differentiation, we also employed the Notch-response-element-lacZ (NRE-lacZ) reporter line to identify progenitor cells (EBs). Our result showed CoA administration remarkably decreased the accumulation of EBs seen in old flies (Fig 2F–2H and 2J). These findings suggest that CoA administration also promotes ISC differentiation of aged flies and thus inhibits intestinal epithelial dysplasia.

## Reduced CoA synthesis in young ISCs results in the deterioration of ISC functionality

Given CoA administration prevents old ISC dysfunction, thus it is reasonable to access whether CoA synthesis decline in ISCs was involved in ISC dysfunction during aging. *dPANK/fbl* initiates the CoA synthesis process, and its suppression leads to a reduction in CoA levels [33] (Fig 3A). Interestingly, analyses from single-cell RNA sequencing demonstrated that the expression levels of *dPANK/fbl* were markedly lower in elderly ISCs than in those from younger individuals (S1A Fig). To further study *dPANK/fbl* expression in aging *Drosophila* ISCs, real-time qPCR was performed on sorted *esg*-GFP+

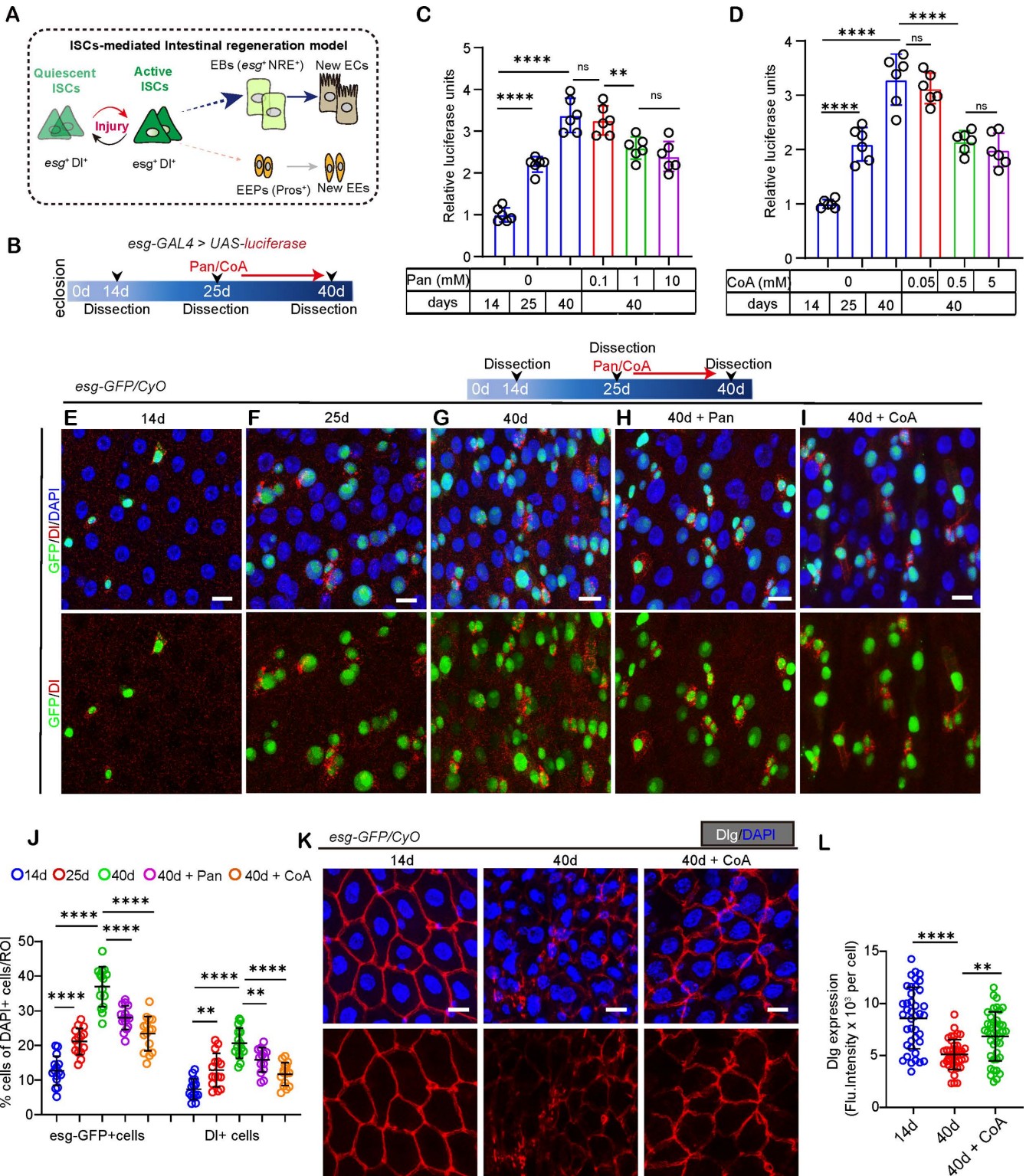

**Fig 1. Orally administered CoA repressed intestinal epithelial dysplasia and barrier dysfunction of aged *Drosophila.*** (A) Schematic representation of the differentiation model for *Drosophila* intestinal stem cells (ISCs) and the lineage-specific markers. (B) Schematic representation of the 'esg>luciferase' workflow used for chemical compound scans in *Drosophila*. (C-D) Assessment of luciferase activity levels in flies of specific age groups following the administration of pantothenate (C) or CoA (D) and each dot represent repeating times. (E-I) Representative immunofluorescence photos

of midguts with GFP (green) and Dl (red) staining from flies with 14d (E), 25d (F), 40d (G), 40d + Pantothenate (H), and 40d + CoA (I). (J) Quantification of the percentage of *esg*-GFP positive cells or Delta (Dl) positive cells to total cells with indicated manipulation with indicated age of midguts in experiments (E-I) and each dot represent one region of interest (ROI) from R4 midguts. (K-L) Representative immunofluorescence photos (K) of midguts with Discs large (Dlg) staining and quantification Dlg expression (L) with indicated manipulation in experiments (K) and each dot represent one ROI from R4 midguts. Scare bar: 10μm. ROI size: $1.5 \times 10^4$ μm². Error bars indicate SDs. Asterisks denote levels of significance: *$p < 0.05$, **$p < 0.01$, ***$p < 0.001$, ****$p < 0.0001$, and ns (not statistically significant) signifies $p > 0.05$. One-way ANOVA with Tukey's multiple comparison test for (C, D, J, L).

cell populations. The results revealed a substantial decline in *dPANK/fbl* mRNA levels in *esg*+ cells with aging (S1B Fig). Consistently, an ELISA assay disclosed a marked decrease in CoA concentrations within the midguts of older *Drosophila* (S1C Fig). Therefore, both the mRNA levels of *dPANK/fbl* and CoA levels in ISCs show a marked decrease with aging. This indicates that the diminished CoA synthesis in older ISCs may contribute to the functional decline observed in ISCs.

To determine if the decline in *dPANK/fbl* expression within aged ISCs leads to the over proliferation of ISCs in the aging midgut, we employed *esg-Gal4*-mediated RNA interference to knock down *dPANK/fbl* expression. *dPANK/fbl* RNAi decreased CoA levels in the midguts of young flies (S1C Fig). Moreover, a decrease in *dPANK/fbl* expression in young flies resulted in an accumulation and hyperproliferation of ISCs (Figs 3B, 3C, S1D–S1H, S2A and S2B), a phenomenon also seen in aged wild-type *Drosophila*. Notably, we discovered that the loss of *dPANK/fbl* in *Drosophila esg*+ cells significantly disrupted the direction of ISC differentiation during gut homeostasis, as indicated by the elevated proportion of NRE+ and pre-EE+ cells in the midguts of young flies (Figs 3D–3H and S1I). More importantly, supplying CoA remarkably corrected the accumulation and hyperproliferation of ISCs induced by *dPANK/fbl* depletion in young flies (Figs 3B, 3C, and S1D–S1H). Moreover, CoA supplementation significantly rescued the ISC differentiation capability of flies with *dPANK/fbl* depletion as indicated by NRE-lacZ staining (Fig 3D–3H) and *esg*-GFP+Pros+ staining (Figs 3E–3H and S1I). However, pantothenate supplementation failed to mitigate the ISC accumulation and disruption in differentiation ability triggered by *dPANK/fbl* deficiency in these young flies (Figs 3B–3H and S1D–S1I).

To elucidate the role of *dPANK/fbl* in age-associated aberrant differentiation of ISCs, we performed Flip-out lineage-tracing RNAi clone analyses. We found that depletion of *dPANK/fbl* resulted in significantly larger clones compared to wild-type (WT) controls (Fig 3I–3K). Immunostaining revealed a marked increase in Dl-positive and pros-positive cells within *dPANK/fbl* RNAi clones following BLM-induced clone induction (ACI), further suggesting disrupted lineage commitment (Fig 3I, 3J, 3L and 3M).

To further assess the impact of *dPANK/fbl* loss on ISC differentiation directionality, we quantified the munber of mature ECs (esg⁻Pros⁻ and polyploid) and EEs (esg⁻Pros⁺). *dPANK/fbl*-depleted midguts exhibited a reduced population of mature ECs but an elevated EE count relative to WT controls (Fig 3N and 3O). Consistent with this, RT-qPCR analysis demonstrated a significant downregulation of EC-specific enzymatic markers in *dPANK/fbl*-deficient flies (Fig 3P), reinforcing the impairment of terminal EC differentiation.

Given the essential role of *dPANK/fbl* in EC production, we hypothesized that its depletion would compromise gut regenerative capacity. Post-injury analysis revealed defective midgut regeneration in *dPANK/fbl* RNAi flies, characterized by persistent accumulation of NRE-lacZ⁺ cells and attenuated reduction of *esg*-GFP⁺ ISC populations compared to pre-injury levels (S1J–S1M Fig). Furthermore, *dPANK/fbl*-deficient flies displayed significantly reduced survival rates under chronic BLM-induced damage, a phenotype partially rescued by CoA supplementation (S1N Fig). Importantly, overexpression of *dPANK/fbl* cDNA in ISCs (Fig 3A and 3J–3M) not only enhanced CoA levels but also fully restored the differentiation capacity of ISC progenies in the midguts of aged flies.

Finally, we explored whether overexpression of CoA in ISCs could delay intestinal aging. Notably, overexpression of *dPANK/fbl* cDNA in ISCs (S1C and S2C–S2G Figs) increased CoA levels, and fully restored the differentiation capacity of ISC progenies in the midguts of aged flies.

Collectively, these findings demonstrate that dPANK/fbl silencing in ISCs disrupts ISC-to-EC differentiation, leading to lineage mis-regulation, impaired tissue repair, and reduced viability upon intestinal stress.

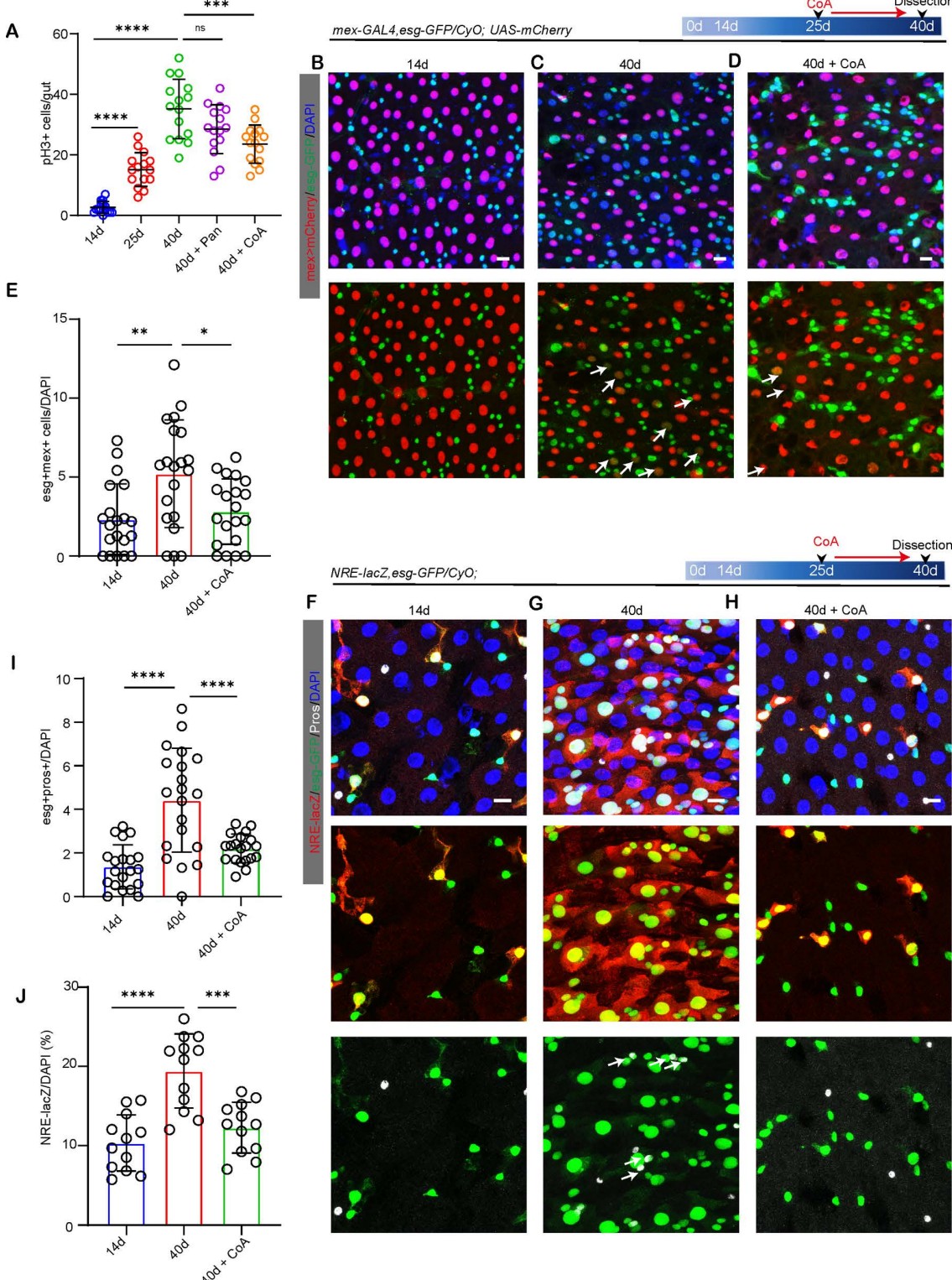

**Fig 2. Exogenous pantothenate/CoA administration prevents ISC functional decline within aged flies.** (A) Qualification pH3 + cells count of midguts from flies with 14d, 25d, 40d, 40d with Pan, and 40d with CoA. Each dot represents one whole midgut. (B-D) Representative photos of *mex-GAL4 > UAS-mCherry* (red) and *esg*-GFP (green) staining from *Drosophila* midgut with 14d, 40d and 40d with CoA. White arrows indicate *mex + esg*-GFP+ cells. (E) Quantification of the percentage of *mex + esg+* cells to total cells within midguts from *Drosophila* in Fig 2B–2D per ROI. Each dot

represents one ROI from one midgut. (F-H) Representative photos of midguts with *esg*-GFP (green), NRE-lacZ (red), and Prospero (Pros,white) staining with indicated treatment. White arrows indicate *Pros+esg*-GFP+ cells. (I-J) Qualification of the percentage of *esg*+Prso+ (I) and NRE-lacZ (J) cells to total cells in Fig 2E–2H per ROI. Each dot represents one ROI from one midgut. Scare bar: 10μm. ROI size: $1.5 \times 10^4$ μm². Error bars indicate SDs. Asterisks denote levels of significance: *p < 0.05, **p < 0.01, ***p < 0.001, ****p < 0.0001, and ns (not statistically significant) signifies p > 0.05. One-way ANOVA with Tukey's multiple comparison test was used.

## CoA counteracts the age-associated deterioration of intestinal functions and prolongs the life expectancy of *Drosophila*

Previous research has demonstrated that the decline in the functionality of ISCs with age leads to notable impairments in the digestive capabilities of *Drosophila*. This includes disrupted gastrointestinal acid-base balance, as well as reductions in both food consumption and waste excretion [34–36]. Given that CoA administration has the potential to restore age-related ISC proliferation, we proceeded to investigate its effects on enhancing the digestive functions of aged flies. As expected, initiating CoA supplementation at a moderately advanced age effectively hindered the progressive deterioration of gastrointestinal acid-base equilibrium (Fig 4A–4C) and contributed to alleviating the diminished food consumption and fecal elimination in elderly flies (Fig 4D and 4E). Furthermore, the addition of CoA enhanced the function of the intestinal barrier in aged flies (Fig 4F). These findings suggest that CoA administration may help avert the decline in gastrointestinal tract function linked to aging in ISCs.

Moreover, reducing *dPANK/fbl* expression in young *Drosophila* resulted in a reduction in gut function that resembled the impairment observed in wild-type older flies (Fig 4G–4J). Importantly, CoA administration completely restored the intestinal function that was compromised due to *dPANK/fbl* depletion in ISCs (Fig 4G–4J).

Given that CoA administration notably impeded the aging of ISCs and alleviated the age-related functional decline of the midgut, it was reasonable to explore whether CoA administration could also extend the lifespan of *Drosophila*. Notably, our observations indicated that lifespan extension of lifelong CoA administration can vary with concentration and was not associated with sex (Figs 4K, 4L, S3A and S3B). And we found 0.5mM CoA was effective on extend lifespan compared with old WT flies. However, a higher CoA (2.5mM) concentration could not further extend lifespan. Moreover, we also investigated whether Pan administration could extend lifespan. We found 1mM Pan was effective to prolong lifespan (S3A and S3B Fig).

## *dPANK/fbl* Function Cell Autonomously to Regulate Intestinal Stem Cell Differentiation in *Drosophila*

Since stem cells fate decision are greatly affected by the surrounding environment niche, we next investigated whether disrupting CoA synthesis in ECs or EEs also influenced ISC proliferation and differentiation. Notably, lineage-specific depletion of *dPANK/fbl* in ECs or EEs failed to recapitulate the ISC-to-EC differentiation defects observed in ISC-targeted knockdown or aged midguts. Unlike ISC-depleted or aged models, EC/EE-specific *dPANK/fbl* loss did not trigger aberrant accumulation of Dl-positive ISCs, *esg*-expressing stem and progenitor cells, or mitotically active pH3+ cells, as evidenced by quantitative analyses (S4A–S4E Fig).

Functional assessments further indicated that targeted knockdown of *dPANK/fbl* in ECs or EEs did not impair intestinal barrier function. Unlike ISC-specific depletion, EC/EE-deficient guts maintained normal Dlg expression levels comparable to young controls (S4F and S4G Fig). Moreover, key physiological parameters—including acid-base homeostasis, feeding behavior, excretion efficiency (S4H–S4L Fig), and epithelial barrier integrity—remained unaffected. Crucially, survival analysis revealed no significant mortality difference between EC/EE-specific *dPANK/fbl* knockdown and wild-type cohorts under homeostatic conditions (S4M Fig).

These findings collectively establish that *dPANK/fbl* regulates ISC differentiation through a cell-autonomous mechanism intrinsic to ISCs, rather than via non-cell-autonomous effects mediated by ECs or EEs.

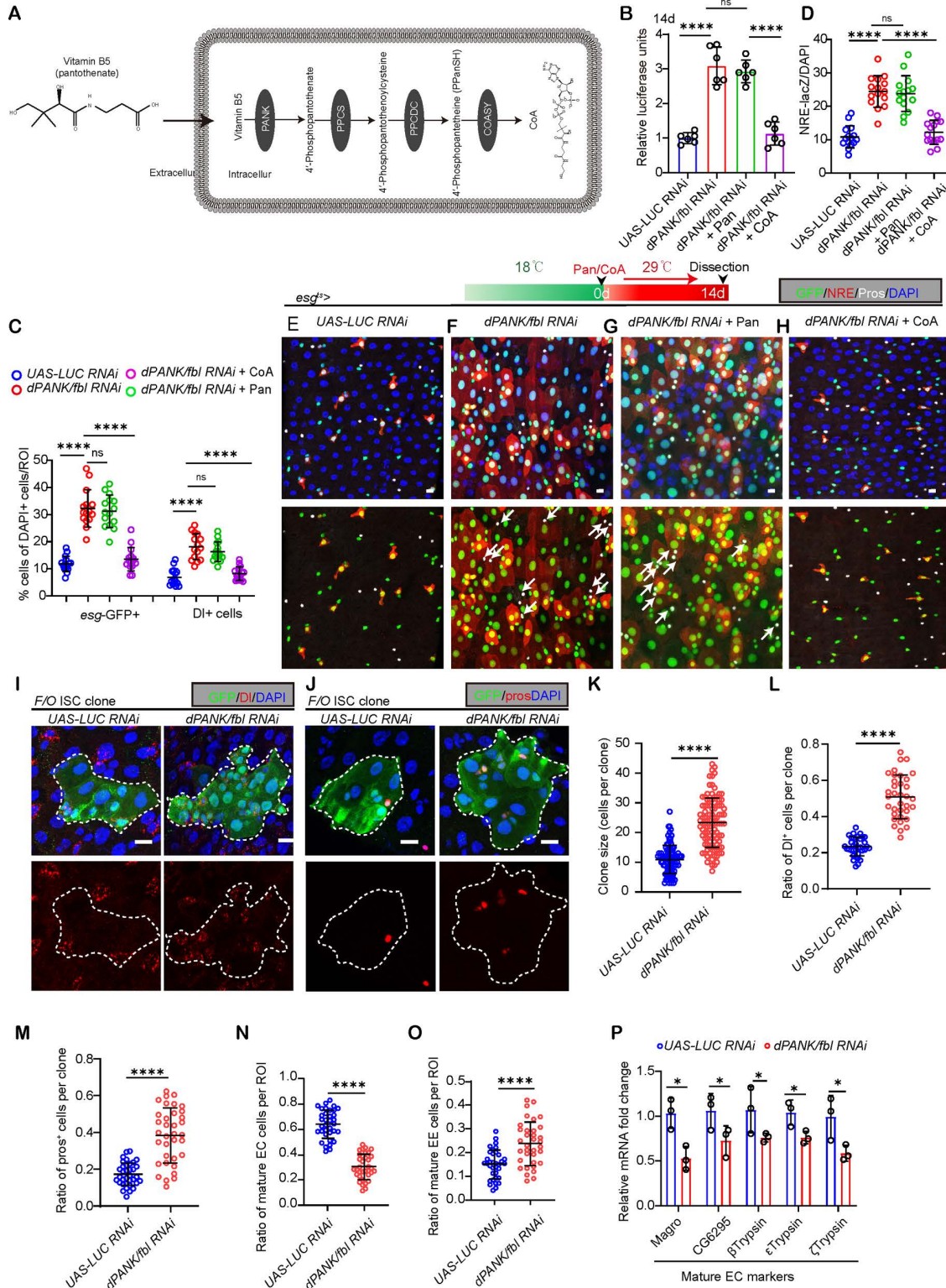

**Fig 3. Reduced CoA synthesis in young ISCs results in the deterioration of ISC functionality.** (A) Diagram of CoA synthetic procedure *in vivo*. (B) Assessment of luciferase activity levels from 14d flies with *UAS-LUC* RNAi (control), *UAS-dPANK/fbl* RNAi and *UAS-dPANK/fbl* RNAi+Pan, and *UAS-dPANK/fbl* RNAi+CoA driven by *esg*-GAL4. Each dot represents repeating times. (C) Quantification the percentage of *esg*-GFP and Dl+ cells to total cells in S1E-S1H Fig. Each dot represents one ROI from one midgut. (D) Quantification of the percentage of NRE-lacZ+ cells to total cells in Fig

3E–3H. Each dot represents one ROI from one midgut. *UAS-LUC* RNAi is used as a control. (E-H) Representative photos with *esg*-GFP (green) and NRE-lacZ+ (red) and Pros (white) staining from flies with indicated administration from 14d flies. UAS-LUC RNAi is used as a control. White arrows indicate Pros+*esg*-GFP+ cells. (I-J) Lineage tracing of ISCs and their progeny using the flip-out (F/O) system from flies treated with BLM and 10 days after colon induction. GFP labeling of ISC lineages and staining with Dl (I) and pros (J). (K-M) Quantification of total number of cells per clone (K), as well as the ratios of Dl-positive cells to total cells (L) and pros-positive (M) cells to total cells, were analyzed in control and *dPANK/fbl* RNAi F/O ISC clones. ISC clones containing three or more cells were quantified. Each dot represents a single clone. (N, O) Quantification the ratio of mature ECs (*esg*-pros- and polyploid) and mature EEs (esg- and pros+) after 10 days driven with *esg*-GAL4.*UAS-LUC RNAi* was used as a control. Each dot represents one ROI from one midgut. (P) Relative mRNA expression of EC-specific genes in midguts of *dPANK/fbl* RNAi flies compared with controls (*UAS-LUC RNAi*) driven by esg-GAL4. Expression of control normalized to 1. n = 3 replicates. Scare bar: 10μm. Error bars indicate SDs. ROI size: $1.5 \times 10^4$ μm². Asterisks denote levels of significance: *p < 0.05, **p < 0.01, ***p < 0.001, ****p < 0.0001, and ns (not statistically significant) signifies p > 0.05. Statistical analyses were conducted using Student's t-tests (K-O), One-way ANOVA with Tukey's multiple comparison test was used (B, D,C,P).

## CoA promotes ISC-to-EC differentiation by inhibiting iron accumulation in ISCs and via a ferroptosis-independent manner

To delve deeper into the mechanisms behind CoA's protective role against intestinal dysplasia, we conducted further research. Initially, we quantified gene expression by exploring RNA sequencing findings from already published research [37]. Employing Gene Set Enrichment Analysis (GSEA), we discovered an enrichment of genes associated with iron homeostasis in cells subjected to CoA treatment (S5A Fig). Since iron accumulation in the brain is a common feature of *PANK2*-associated neurodegeneration [38]. We next studied whether *dPANK/fbl* uses the same mechanism to regulate ISC function.

Our findings indicated that a deficiency in *dPANK/fbl* resulted in a marked elevation of iron levels in ISCs when contrasted with the control group of young flies, as indicated by a reporter line (UAS-Fer1HCH, A crucial component of the iron storage protein complex) and RhoNox-1 staining (Figs 5A, 5B, S5B and S5C). More importantly, Supplementation with CoA could significantly counteract the iron surplus phenotype induced by the absence of *dPANK/fbl* in ISCs of young *Drosophila*. (S5B and S5C Fig). To test whether inhibiting iron overload could rescue the ISC accumulation and block ISC differentiation phenotype in *dPANK/fbl* knockdown flies, we treated *esg-GAL4, UAS-GFP; UAS-dPANK/fbl* RNAi flie lines with bathophenanthrolinedisulfonic acid (BPS, iron chelator) to reduce iron levels in ISCs. We observed a significant rescue ISC dysfunction phenotype caused by *dPANK/fbl* knockdown (Fig 5C and 5D).

To test whether iron overload is involved in CoA's rescue effect on the function of aged ISCs, we performed RhoNox-1 staining, we found that iron content was also increased in ISCs of aging flies compared with young flies (Fig 5E and 5F). Moreover, these iron overload phenotypes could be significantly rescued by CoA administration (Fig 5E and 5F). Importantly, iron chelation with BPS could significantly rescue the ISC dysfunction phenotype of old flies (Fig 5G–5I and 5L). However, FAC (iron salt ferric ammonium citrate, an iron supplement) administration negated the benefits of CoA treatment in old flies (Fig 5G–5L). These findings suggest that iron acts subsequent to CoA in the regulation of ISC function.

Previous research has established that iron overload triggers ferroptosis, which in turn exacerbates intestinal diseases [39]. To determine whether ferroptosis contributes to the functional decline of ISCs following *dPANK/fbl* depletion, we initially examined the expression of genes associated with ferroptosis. Intriguingly, we observed upregulation of both pro- and anti-ferroptotic genes in *dPANK/fbl*-depleted ISCs, presenting a paradoxical regulatory pattern (S5D Fig). We further assessed lipid peroxidation, a key hallmark of ferroptosis. Our results indicated that ISCs from *dPANK/fbl*-depleted flies exhibited lipid peroxidation levels similar to those in control ISCs (S5E and S5F Fig). Complementary quantification of 4-hydroxynonenal (4-HNE), an end-product of lipid peroxidation, further confirmed comparable oxidative lipid damage in both experimental groups (S5G and S5H Fig). This consistent absence of ferroptosis biomarkers strongly suggests that *dPANK/fbl* depletion disrupts ISC-to-EC differentiation through ferroptosis-independent mechanisms.

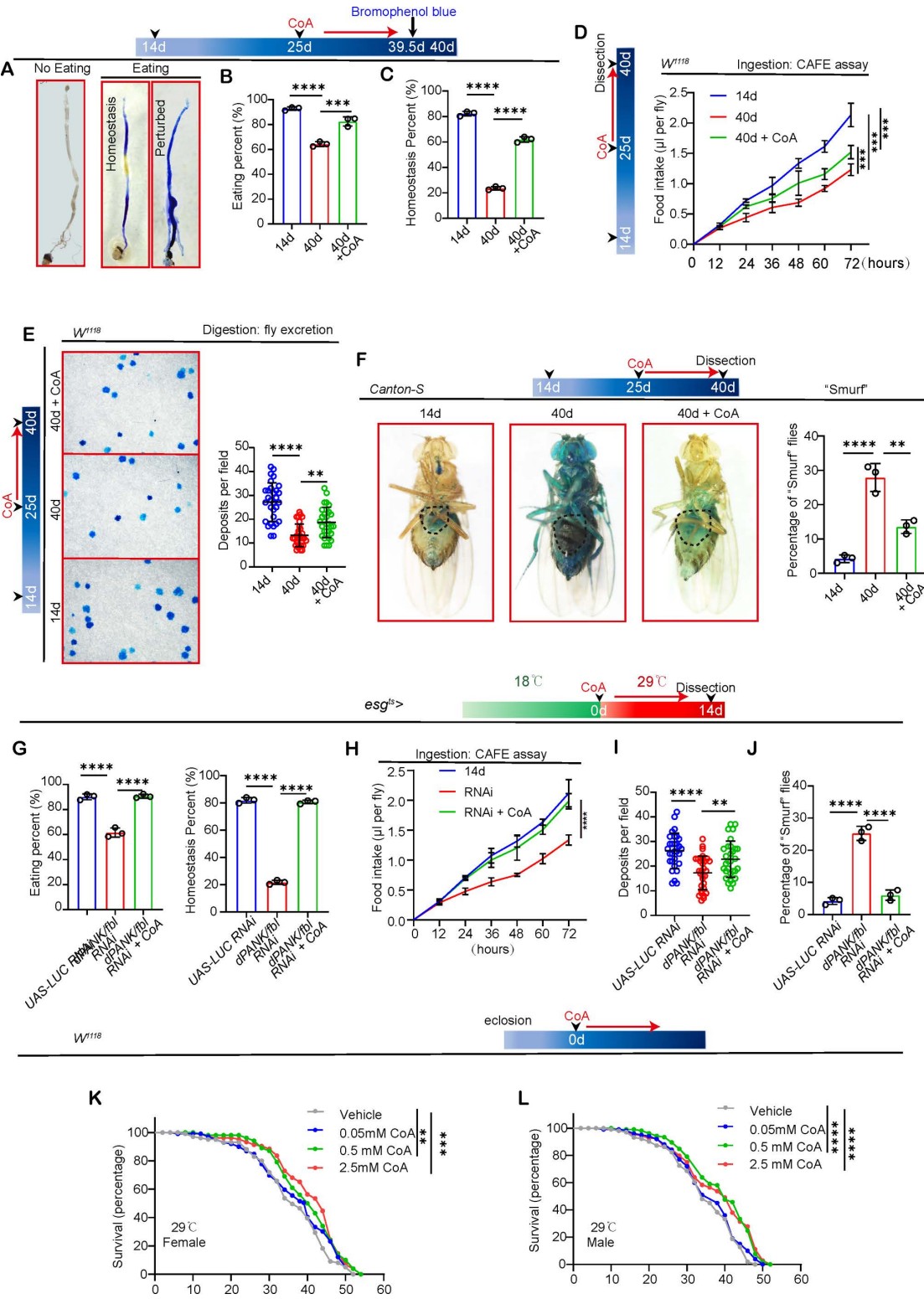

**Fig 4. CoA counteracts the age-associated deterioration of intestinal functions and prolongs the life expectancy of *Drosophila*.** (A-C) Representative photos (A) and corresponding measurements (B-C) of intestinal acid-base equilibrium in *Drosophila* at the 14d (control), 40d, and 40d+CoA of flies. Each dot represents replicate times. (D) Assessment of dietary consumption in *Drosophila* through the CAFE assay from flies with 14d (control), 40d, and 40d+CoA. Error bars show the SD of three independent experiments. (E) Presented are images of deposits along with the quantification of

their numbers from 14d (control), 40d, and 40d + CoA of flies. Each dot represents one quantified field from 20 *Drosophila*. (F) Images and numerical data depicting the proportion of 'Smurf' phenotype flies at 14d (control), 40d, and CoA-supplemented 40-day flies post consumption of a non-absorbable food dye. Each dot represents one replicate times. (G) Quantification of the eating percent and homeostasis percent from flies with *UAS-LUC* RNAi (control), *UAS-dPANK/fbl* RNAi and *UAS-dPANK/fbl* RNAi + CoA driven by *esg*-GAL4. Each dot represents one replicate times. (H) Assessment of dietary consumption in *Drosophila* through the CAFE assay from 14d flies with *UAS-LUC* RNAi (control), *UAS-dPANK/fbl* RNAi and *UAS-dPANK/fbl* RNAi + CoA. Error bars show the SD of three independent experiments. **(I)** Quantification of deposits number from 14d flies with *UAS-LUC* RNAi (control), *UAS-dPANK/fbl* RNAi and *UAS-dPANK/fbl* RNAi + CoA. Each dot represents one quantified field from 20 *Drosophila*. (J) Quantification of percentage of "Smurf" of indicated treatment. Each dot represents one replicate times. (K, L) Percentage survival rates of female flies (K) and male flies (L) and using Canton-S as a wild-type, with CoA supplementation and without it starting at 1-2-day old are presented. The numbers of quantified Drosophila: 100 for each group. Three independent experiments were conducted. The findings are based on three separate experimental trials. Each dot represents one replicate times. Error bars indicate SDs. Asterisks denote levels of significance: $*p < 0.05$, $**p < 0.01$, $***p < 0.001$, $****p < 0.0001$, and ns (not statistically significant) signifies $p > 0.05$. One-way ANOVA with Tukey's multiple comparison test was used unless two-way ANOVA with Tukey's multiple comparison test was conducted for (D, H). Survival curves were analyzed using the log-rank (Mantel-Cox) test (K, L).

## CoA administration prevents ISC dysfunction through antioxidative ability and protects ISCs from stress-induced DNA damage

As excess iron can accelerate reactive oxygen species (ROS) generation, we next detected ISC ROS activity using a reporter line, *GsTD1-GFP* and DHE staining to assess its role in this phenotype. First, we found that FAC treatment could also significantly elevate ROS activity in young ISCs compared with vehicle-treated flies (S6A and S6B Fig). Importantly, ISCs showed higher ROS activity in *Drosophila* with *dPANK/fbl* depletion compared with flies with *LUC* depletion (Figs 6A, 6B, S6A and S6B). However, these ROS elevation phenotypes of *dPANK/fbl* depletion flies could be significantly rescued by CoA administration (S6A and S6B Fig). Thus, these results indicate that of *dPANK/fbl* depletion could elevate ROS activity in ISCs.

Our findings indicate a link between the absence of *dPANK/fbl* in ISCs and the buildup of ROS. To determine if a causal relationship exists, we investigated whether the removal of ROS could counteract the senescence phenotype observed in *Drosophila* ISCs when *dPANK/fbl* is depleted. By dietary administration, we examined substances recognized for their antioxidant characteristics, such as N-acetyl-L-cysteine (NAC). Findings indicated that NAC could significantly rescue ISCs' senescence phenotype from flies with *dPANK/fbl* depletion in ISCs (Fig 6C–6G). Though NAC consumed in food could be readily absorbed by the intestinal cells, we could not exclude protective effects of NAC in other organs, since these compounds could also penetrate other organs. To isolate the effects on the gastrointestinal tract, we specifically targeted the expression of the antioxidant enzyme Catalases (*CAT*) within ISCs using *esg-GAL4.* Consistent with the rescue effect of NAC administration, overexpression of *CAT* in ISCs could significantly rescue the senescence phenotype of ISCs with *dPANK/fbl* depletion (Fig 6C–6G). With age, we also found CoA administration could reduce ROS activity in old ISCs (S6C and S6D Fig). Thus, our results indicate *dPANK/fbl* promotes ISC function by inhibiting ROS activity.

Given that excess ROS are recognized for inducing DNA damage and that the accumulation of damage from environmental stress is widely considered a major factor in organismal aging, we investigated whether CoA administration could protect ISCs from such stress-induced damage. To test this hypothesis, we exposed ISCs to various stress factors, including BLM treatment (a DNA damage inducer) and aging. We monitored the phosphorylated form of γH2AvD, a recognized marker of double-strand DNA breaks, in response to these stressors. We first found iron administration using FAC indeed induced γH2AvD elevation in ISCs (Fig 6H, 6I and 6N). We also observed that the nuclear intensity of γH2AvD significantly increased in ISCs with *dPANK/fbl* depletion (Fig 6H, 6J, and 6N). However, this phenotype was mitigated by CoA administration (Fig 6H, 6J, 6K and 6N). Furthermore, γH2AvD intensity was elevated in ISCs from aged flies (Fig 6H, 6I and 6K), and similarly, CoA treatment reduced the heightened γH2AvD intensity in these old ISCs (Fig 6E, 6L, 6M and 6N). These findings suggest that CoA protects ISCs from stress-induced damage, reducing the damage induced by both DNA-damaging agents and the natural aging process.

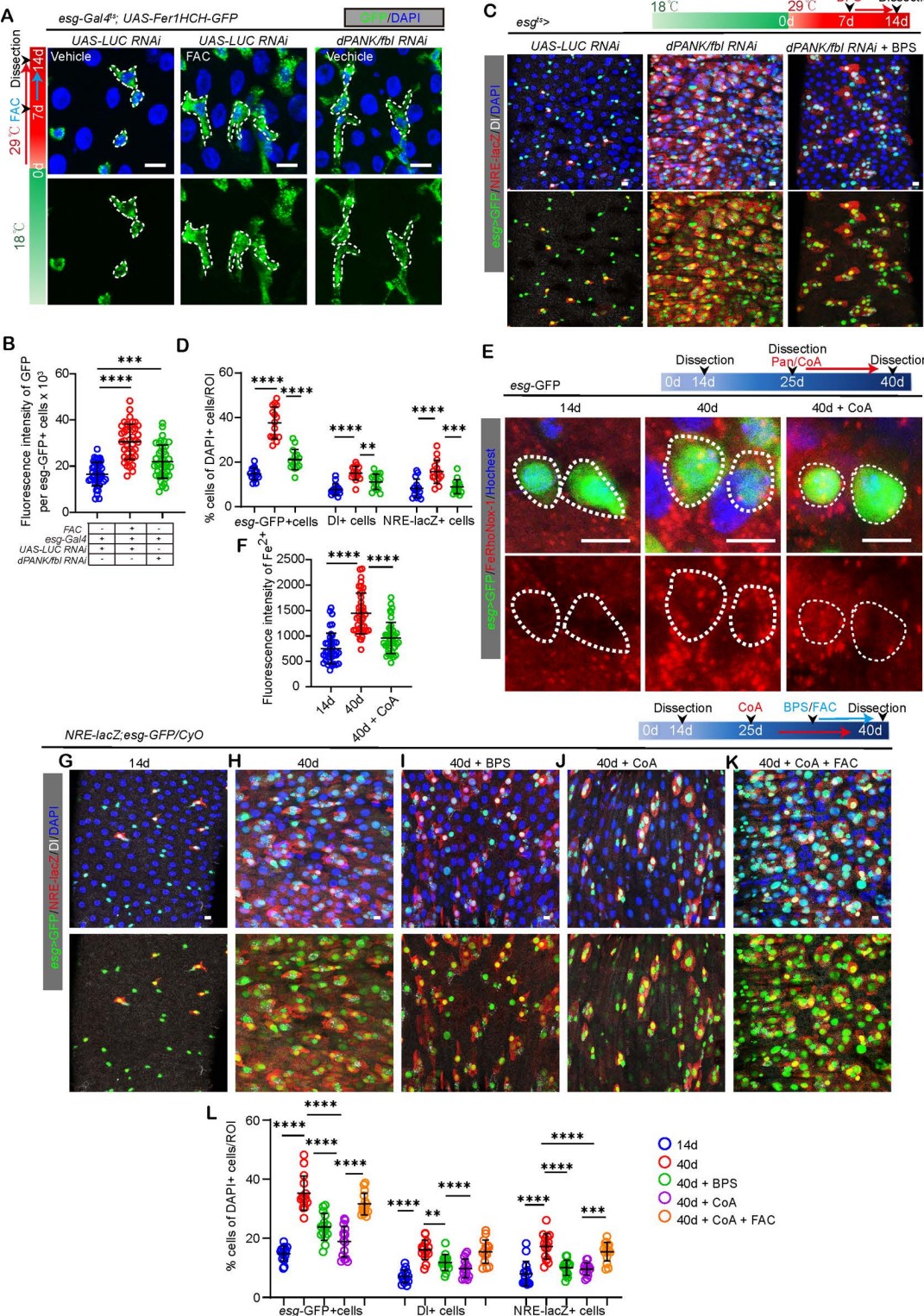

**Fig 5. CoA promotes ISC differentiation by inhibiting iron accumulation in ISCs.** (A) Representative photos from midguts with esg-GAL4 > UAS-Fer1HCH-GFP staining (iron level) from 14d flies with *UAS-LUC* RNAi (control), and *UAS-dPANK/fbl* RNAi. (B) Quantification of fluorescence intensity of Fer1HCH-GFP per ISCs from flies with indicated treatment. Each dot represents mean fluorescence intensity randomly chosen from 10 *esg*-GFP+ cells per ROI from 20 midguts. (C) Representative photos with *esg*-GFP (green), DI (white), and NRE-lacZ (red) staining from midguts of 14d flies with

*LUC* RNAi (control), *UAS-dPANK/fbl* RNAi and *UAS-dPANK/fbl* RNAi + BPS (an iron chelator). (D) Quantification of percentage of *esg*-GFP positive cells, DI positive cells, and NRE-lacZ positive cells to total cells in Fig 5C. Each dot represents one midgut. (E) Representative photos from midguts with *esg*-GFP (green) and RhoNox-1 staining (red) from 14d (control), 40d, and 40d + CoA of flies. (F) Quantification of iron fluorescence intensity per ISCs from flies with associated age and drug administration in (E). Each dot represents mean fluorescence intensity randomly chosen from 10 esg-GFP+ cells per ROI from 20 midguts. (G-K) Representative photos with *esg*-GFP, DI, and NRE-lacZ staining from midguts of 14d (control), 40d, and 40d + BPS, 40d + CoA and 40d + CoA + FAC (iron salt ferric ammonium citrate, an iron supplement) flies. **(L)** The count of percentage of *esg*-GFP, DI, and NRE-lacZ positive cells to totcal cells per ROI is in Fig 5G–5K. Each dot represents one midgut. Scare bar: 5µm (E), 10µm (A, C, G-K). Error bars indicate SDs. ROI size: $1.5 \times 10^4$ µm². Asterisks denote levels of significance: *$p < 0.05$, **$p < 0.01$, ***$p < 0.001$, ****$p < 0.0001$, and ns (not statistically significant) signifies $p > 0.05$. One-way ANOVA with Tukey's multiple comparison test was used.

## Discussion

The role of metabolites in regulating ISC function is increasingly recognized [40,41]. Our study aimed to discover a novel metabolic pathway contributing to ISC senescence. We found a downregulation of the pantothenate/CoA pathway in ISCs from older flies. Notably, exogenous CoA supplementation, a key metabolite, could metabolically reconfigure aged ISCs, reducing iron levels and restoring a youthful phenotype (Fig 6O).

Disruptions in the CoA biosynthetic pathway can have detrimental effects, and genetic mutations in enzymes involved in CoA synthesis can lead to inherited human diseases [14,38]. These conditions include pantothenate kinase-associated neurodegeneration and various heart diseases. These findings highlight the necessity of tight regulation of the CoA synthesis pathway for the survival of living organisms. However, the roles of CoA in aging, particularly its impact on stem cell regulation, are largely unexplored. Our research underscores the significant role of CoA in preventing age-related decline in stem cell function. Mechanistically, our research reveals that CoA reduces excessive ISC senescence by lowering iron levels within ISCs.

In this study, we identified a strong correlation between iron accumulation and CoA deficiencies in ISCs. Our results align with prior research indicating a link between CoA deficiencies and brain iron accumulation [42,43]. Consequently, these results underscore the conservation of this mechanism. However, the specific mechanism by which CoA inhibits iron levels in ISCs remains to be elucidated. As mentioned earlier, our study didn't clarify how iron accumulates in ISCs when *dPANK/fbl* is lost. But we can look at research on PANK2-linked neurodegenerative diseases for clues. A key idea is that PANK2 deficiency leads to enzyme shortage, causing CoA biosynthesis substrates to pile up and increasing N-pantothenyl cysteine and free cysteine [14,44]. Cysteine can bind iron, possibly causing iron accumulation. This might explain the iron buildup in *dPANK/fbl*-deficient ISCs. Also, PANK2's mitochondrial location has spurred research into mitochondrial issues in PKAN. *Santambrogio* et al. [42] found tubulin acetylation/phosphorylation defects in PKAN astrocytes, likely disrupting vesicle dynamics and iron delivery to mitochondria, leading to iron misplacement. Cytosolic iron overload and mitochondrial iron shortage may cause mitochondrial problems. We will study vesicle dynamics and iron movement in ISCs in future work. Given the shared regulatory pathways among various somatic stem cells, it is reasonable to hypothesize that CoA could act via a universal mechanism in different adult stem cell types.

Previous research has documented iron homeostasis disruptions in a significant portion of the aging population [45,46]. However, the specific levels of iron accumulation in aged ISCs and its exact role in regulating ISC senescence are yet to be fully understood. Our data show that increased iron levels in aging ISCs induce senescence, likely by enhancing reactive ROS activity and causing subsequent DNA damage. It is well-established that ROS can activate the c-Jun N-terminal kinase (JNK) pathway. The sustained activation of JNK in stem cells is known to cause over-proliferation of ISCs and differentiation defects in their progeny [47]. We think a similar process might be behind ISC iron accumulation with *dPANK/fbl* loss. However, as mentioned earlier, lowering iron levels can trigger various changes, and any of these could be the main cause of the observed phenotypic shifts. This is understandable, considering iron's role as a cofactor for vital parts of the electron transport chain and many other enzymes (like catalase and lipoxygenases), as well as for components of the iron regulatory system. We suggest that the phenotypic outcomes in ISCs result from multiple interacting factors. Further

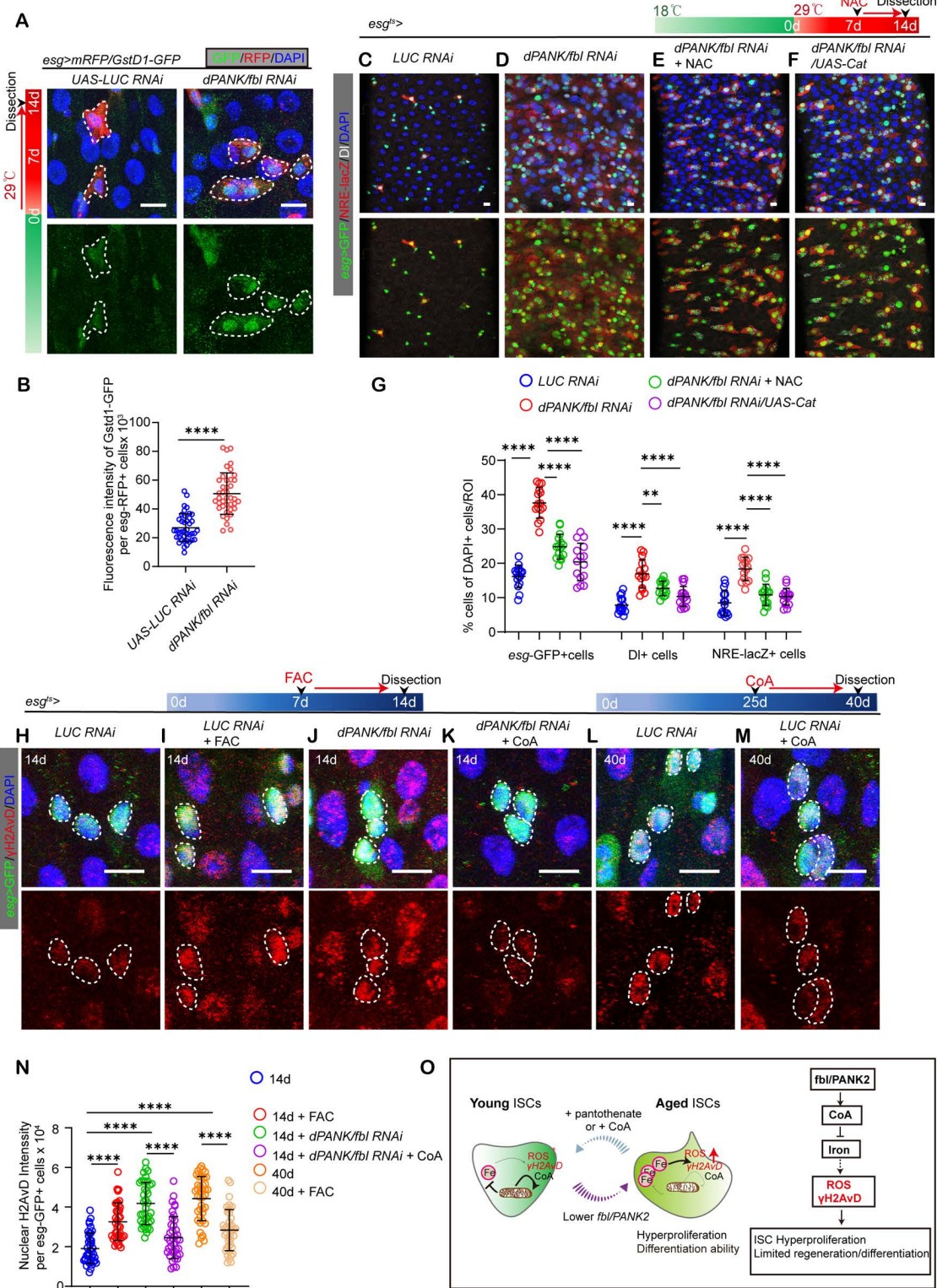

**Fig 6. CoA administration prevents ISC dysfunction through antioxidative ability and protects ISCs from stress-induced DNA damage.** (A) Representative photos of midguts with *GstD1*-GFP (green) and *esg > mRFP* staining (red) from 14d flies with *UAS-LUC* RNAi (control), and *UAS-dPANK/fbl* RNAi. (B) Quantification of *GstD1*-GFP fluorescence intensity per ISCs from flies with indicated administration in **(A)**. Each dot represents mean fluorescence intensity randomly chosen from 10 esg-GFP+ cells per ROI and total of 19 midguts. (C-F) Typical images from midguts stained for

esg-GFP (green), Dl (white), and NRE-lacZ (red) from 14d flies with *UAS-LUC* RNAi (control), *UAS-dPANK/fbl* RNAi and *UAS-dPANK/fbl* RNAi + NAC, *UAS-dPANK/fbl* RNAi + *UAS-Cat*. (G) Measurement of the ratio of esg-GFP, Dl, and NRE-lacZ counts to total cells in the Fig 6C–6F . Each dot represents one midgut. (H-M) Typical images of midguts stained for esg-GFP (green) and γH2AvD (white), with the specified treatments labeled. (N) Quantitative assessment of the mean γH2AvD fluorescence signal in esg-GFP-positive cells. Each dot represents mean fluorescence intensity randomly chosen from 10 esg-GFP+ cells per ROI and total of 16 midguts are quantified. (L) A summary model showed a decrease in activity within the pantothenate/CoA route in ISCs from older flies. Notably, the addition of CoA, a crucial metabolite, exogenously, was capable of metabolically reprogramming these aged ISCs. This reprogramming resulted in a reduction of iron levels within the ISCs, effectively rejuvenating them to a state that resembled their more youthful state. Scale bar: 10μm (A, C), 5μm (E-J). Error bars indicate SDs. ROI size: $1.5 \times 10^4$ μm$^2$. Asterisks denote levels of significance: *$p < 0.05$, **$p < 0.01$, ***$p < 0.001$, ****$p < 0.0001$, and ns (not statistically significant) signifies $p > 0.05$. One-way ANOVA with Tukey's multiple comparison test was used.

research is needed to clarify how these molecular mechanisms specifically contribute to the observed phenotypes. More-over, our results showed supplementing flies with the iron chelator, BPS, significantly reverses the senescence phenotype in flies with *dPANK/fbl* depletion or those that are aging. Additionally, overexpressing CAT or administering NAC to reduce ROS levels significantly reversed the ISC senescence phenotype.

In summary, our findings highlight the importance of the pantothenate/CoA pathway in rejuvenating ISCs in aging *Drosophila*. Together, these results establish a foundation for future research on the therapeutic potential of modulating this pathway to enhance ISC function and extend the lifespan of aging individuals.

## Materials and methods

### *Drosophila* rearing and maintenance

All *Drosophila melanogaster* is nurtured on a conventional diet comprising cornmeal and yeast, with the formulation for 1 liter of medium being: 80 g of sucrose, 50 g of cornmeal, 20 g of glucose, 5 g of agar, 30 mL of propionic acid and 18.75 g of yeast [48]. Unless otherwise indicated, the flies are reared at 25°C under a standard photoperiod. To inhibit the GAL4-driven system, hybrid strains are kept at 18°C for the induction of temperature-response GAL4-mediated RNA interference or gene overproduction. After emergence or at a specified age following emergence, adult flies are transferred to 29°C to trigger the GAL4 system, thus inducing the desired RNAi or gene expression. After a specified period of culture at 29°C, the eclosed flies are subjected to midgut dissection for immunofluorescence analysis. Unless specifically mentioned, experiments concerning the midgut of *Drosophila melanogaster* are conducted using mated female flies.

### Fly strains and genetics

For the present study, the wild-type controls were the $w^{1118}$ (BDSC3605) and *Canton-S* (BDSC64349) alleles. No discernible differences [5] were noted between the esg-GFP/CyO and $w^{1118}$ flies, which is why the esg-GFP/CyO strain is also occasionally used as a control in our experiments. The UAS-luciferase strains (BDSC# 61687), *and LUC*$^{RNAi}$ (BDSC# 31603) were purchased from the Bloomington Drosophila Stock Center (BDSC). *dPANK/fbl* RNAi strains were purchased from the Vienna Drosophila RNAi Center (VDRC, Vienna, Austria) (v101437). The *esg-GAL4,* UAS-GFP, tub-GAL80$^{ts}$/*CyO,* and NRE-lacZ were kindly provided by Benjamin Ohlstein (University of Texas Southwestern Medical Center). UAS-dPANK/fbl was kindly provided by Zhou Bing (Tsinghua University). *GsTD1-GFP* was kindly provided by Guo Zheng (Huazhong University of Science and Technology). The *esg-GFP/CyO* was kindly provided by Allan Spradling (Carnegie Institution for Science). The *mex-GAL4, esg-GFP* lines were made in our lab using standard combined or recombined methods. The Additional *Drosophila* strains employed in this research are delineated within the figures, their respective legends, and the body of the text. To generate RNAi clones, we employed the FRT "flip-out (F/O)" method. GFP-expressing cells were produced by crossing *hsFLP; Act.FRT-CD2-FRT.GAL4, UAS-GFP/CyO* flies (a gift from Allan Spradling) with dPANK/fbl-RNAi,or control flies. The crosses were maintained at 25°C. To induce clones, adult flies aged 2–3 days post-eclosion were subjected to a 50-minute heat shock in a 37°C water bath. Following the heat shock, flies

were transferred to 29°C and maintained on BLM-containing food for 24 hours between 2 and 3 days after clone induction (ACI). The flies were then dissected and analyzed at 10 days ACI. The fly lines were repeatedly crossed back into the $w^{1118}$ background over six generations, and sibling populations were obtained from the crosse process before being used in the experiment.

## Intestinal sample collection and CoA analysis

Midgut tissues from 10-day-old (juvenile) and 40-day-old (senescent) flies were harvested for subsequent metabolomic analysis. Post-dissection, the gut tissues were rapidly cryopreserved in liquid nitrogen. Concentrations of CoA were measured using assay kits from Sigma, following the protocol provided by the manufacturer.

## Fluorescence-activated cell sorting and quantitative gene expression analysis

A hundred dissected female midgut samples were immersed in cooled DEPC-PBS and treated with elastase (1 mg/ml, Sigma, E0258) at 25°C for an hour, with periodic mixing every quarter hour. The enzymatic reaction was halted by centrifugation at 4°C using 400 g for 20 minutes, followed by resuspension using cold DEPC-PBS and filtration through a 70-micron mesh. The GFP+ cells from *esg*-GFP-marked flies were isolated using a FACS Aria II (BD Biosciences), with the $w^{1118}$ strain serving as a control for fluorescence gating. This sorting was consistently performed to yield approximately 40,000 GFP+ cells per biological replicate across three iterations. RNA extraction was conducted utilizing the Arcturus PicoPure kit (Applied Biosystems), adhering to the vendor's instructions. cDNA synthesis was achieved with the PrimeScript RT kit (TaKaRa). The reverse transcription was performed using all RNA, primed with oligo dT primers. Then synthesized cDNA was diluted and underwent quantitative PCR employing SYBR Green (Genestar) on a QuantStudio 5 (Thermo Fisher Scientific), with each sample assayed in triplicate. Then $2^{-\Delta\Delta Ct}$ approach was employed to determine RT-qPCR results, normalizing to the housekeeping gene Rp49, with control samples set to a normalized expression value of 1.

qPCR Primer sequences utilized in this research are shown:

| *Magro*-F | AGCACGGAACCTCCTTCATCTTCA | *Magro*-R | TAGTGGATGCCCTGGTTGCTAGA |
|---|---|---|---|
| *CG6295*-F | TTCATCAACTACGGTGTCCGCGAT | *CG6295*-R | CAGCTGGCCGTTCTTCACATTCTT |
| *βTrypsin*-F | TGGAGCTCTTGATCTGGTTTCCGT | *βTrypsin*-R | TCGTCGCCAAGGTTTCCTCTTTCA |
| *εTrypsin*-F | TAGACACCGGGATACCTCACATC | *εTrypsin*-R | TCCAGCATTCGCGAGATCCGTATT |
| *ζTrypsin*-F | TTGTTCCAAGGCCAGCTTAATGG | *ζTrypsin*-R | TTTCCCTGCGCTACAAGGGTATCA |
| *Cncc*-F | GAATGACCGCCGATCTCTTGG | *Cncc*-R | GGAGCCCATCGAACTGACA |
| *Cat*-F | AGAACTACTTTGCTGAGGTGGA | *Cat*-R | TTCACCTTGTACGGGCAGTT |
| t*sf1*-F | CCTTCCGTTACGAGGGCATTA | t*sf1*-R | CAGCTTGGTGATGGGGATCTT |
| *Fer1HCH*-F | ATGGTGAAACTAATTGCTAGC | *Fer1HCH*-R | TCAGATCGCTGACTCCCTC |
| *Gclc*-F | CAGCTCAATGAGAAGGAACTGG | *Gclc*-R | GTGTGCCCTCAATCATGTAGG |
| *HO1*-F | GGGAACAATAGCTGCGCAAA | *HO1*-R | CGTCATCAGAAAGGGCAAGTG |
| *Gtpx*-F | AGCTGACGGATCTAAAGGAGAA | *Gtpx*-R | GCATCTGGGACCCAAACTGATT |
| *Caly*-F | ACCCAGGTCTCTTCACACTG | *Caly*-R | TCTGTAGGTCGTACACCTCCT |
| *Acsl*-F | CTTACACGTCGAAGCGTTGTC | *Acsl*-R | GTCTTCCACTTGTAGTCACCG |
| *dPANK/fbl*-F | TTCTCTTCGCCGATCTGCATA | *dPANK/fbl*-R | GAACTGCTGCTTTTCGCTTTTTA |
| *rp49*-F | GCTAAGCTGTCGCACAAATG | *rp49*-R | GTTCGATCCGTAACCGATGT |

## Immunofluorescence microscopy analysis of *Drosophila* tissue

Adult midguts from fruit flies are meticulously dissected in chilled phosphate-buffered saline (PBS) and subsequently, 4% EM-grade paraformaldehyde solution was used to fix midguts. The fixation solution contains 100 mM sodium phosphate, 25 mM KCl, 20 mM MgSO4, 4 mM Na2HPO4, and 1 mM MgCl2, adjusted to pH 7.4. Samples were preserved in the

solution for a duration of 30–60 minutes. Following fixation, the samples are extensively washed using a PBS buffer solution containing 0.3% Triton X-100, repeated three to five times for 10 minutes each to ensure thorough removal of fixative.

The samples are then incubated in a blocking solution with 0.5% bovine serum albumin (BSA) for 30 minutes to reduce non-specific antibody binding. Primary antibodies, which have been pre-adsorbed against embryos fixed in 4% paraformaldehyde to enhance specificity, are applied and kept at 4°C for an entire night. Then the midguts undergo three rounds of washing to remove unbound antibodies. Subsequently, the midgut samples are incubated with fluorophore-conjugated secondary antibodies at a dilution of 1:1,000, along with DAPI (4,6-diamidino-2-phenylindole) at a 1 µg/ml working concentration for nuclear staining. This incubation is carried out at ambient temperature for 2 hours, allowing for the antibodies to bind to their respective antigens. Following the secondary antibody incubation, the samples are washed again to remove any unbound secondary antibodies. The following antibodies were used: Chicken polyclonal anti-GFP (1:1000, Abcam, ab13970); Mouse anti-Delta (1:100, Developmental Studies Hybridoma Bank (DSHB), C594.9B); Rabbit anti-phosphoHistone H3 (Ser10) (1:1000, Sigma, 06–570); Chicken anti-β-Galactosidase (1:1000, Abcam, ab9361); Rat Monoclonal anti-mCherry (1:1000, Invitrogen, M11217); anti-discs large (1:100, DSHB, 4F3); DHE (3 µM, Invitrogen, D11347); γH2AvD (1:1000, Rockland, 600401914); RhoNox-1 (1µM, MCE, HY-D1533). BODIPY 581/591 C11 (1:1000, Beyotime, S0043S). 4-Hydroxynonenal Rabbit mAb (1:200, ABclonal, A26085). The images of immunofluorescence are obtained using a Leica TCS-SP8 confocal microscope, with consistent settings applied across all experimental groups to ensure comparability. The assembled images are processed using Adobe Photoshop and Adobe Illustrator CS3 for clarity and presentation. For quantitative analysis, the number of cells within the midgut is determined using a Leica DM6-B microscope, ensuring an accurate count for each experimental condition.

### DHE, BODIPY 581/591 C11 and RhoNox-1 staining

Freshly dissected midgut was immersed in working solution for 30mins, and washed with PBS 3 times. Hoechst was used to stain the nucleus and was detected immediately using confocal microscopy.

### Assessment of luciferase activity

The luminescence of the firefly luciferase was quantified utilizing the Firefly Luciferase Assay Kit (Biyuntian Biotech, China, RG051S). A batch of fifteen female mature *Drosophila* midguts was excised in chilled PBS and swiftly frozen with liquid nitrogen. The samples were lysed in the 50 µl Luciferase Assay Buffer supplied within the kit. Following this, the lysates were spun down with 13,000 g at 4°C for 10 minutes to separate the soluble fraction for analysis. Collection of samples was carried out over multiple days, with the extracts being preserved at -80°C until a total of six replicates per genotype were assembled. The determination of luciferase activity was conducted strictly in line with the instructions from the kit.

### Natural small molecule compounds screening

We conducted small molecule screening utilizing the "*esg*-luciferase" reporter system as described above. Our screening set included Alpha-lipoic acid (Aladdin, D118666), N-acetylcysteine amide (AD4, Aladdin, N170064), taurine, CoA and spermidine (Aladdin, S107071). These compounds were dissolved in DMSO, a procedure reported in our prior research [10,11]. Subsequently, they were added into the regular food medium. The flies were collected and equally distributed among food vials with these natural small molecule substances.

### Bromophenol blue treatment

The assay was performed according to established protocols. 100 µL volume of bromophenol blue sodium solution (final concentration 2%, Sigma, B5525), serving as a pH indicator, was introduced into the food vials. Subsequently,

the food was punctured 4–6 times using a 200µl pipette tip to facilitate absorption of the blue dye. After a 12-hour incubation period, images were promptly captured immediately post-dissection of each midgut. This approach ensures a reliable assessment of the pH changes within the digestive tract, as indicated by the color shift of the bromophenol blue.

**Cafe assay and fly excretion assessment**

The Cafe assay and the quantification of *Drosophila* fecal output were executed following established methodologies [36,49].

**Drug treatments**

The aqueous solutions of pantothenic acid and CoA were integrated into the regular dietary medium used for culturing. *Drosophila* specimens were gathered and uniformly allocated to vials of medium supplemented with these compounds.

***Drosophila* survival tests**

In the longevity assessment, a cohort of one hundred genetically identical female *Drosophila*, 1–2 days post-eclosion, was amassed and equitably apportioned across four feeding tubes. A pair of these tubes were filled with the customary nutritional medium, in contrast to the other pair which was enriched with pantothenic acid and Coenzyme A. To facilitate reproductive processes, a complement of ten male counterparts, also 1–2 days old, was incorporated into each tube. The vitality of the female population was monitored biweekly. The experiment was replicated thrice to reinforce the validity of the findings.

**Quantitative and statistical analysis**

Data are presented as averages with their standard deviations (SD), with figures in this research reflecting outcomes from a minimum of three separate experimental runs, unless specified differently in the methodology or figure captions. The threshold for statistical significance is set using a two-tailed Student's t-test unless otherwise indicated in the legends. Significance levels are explicitly designated within the manuscript or the associated figure captions. A P-value below 0.05 is acknowledged for all analyses as indicative of statistical significance.

**Fluorescence intensity statistics**

All immunofluorescence images were acquired using a Leica TCS-SP8 confocal microscope. For consistency, confocal stacks were captured under identical settings for each experimental set. Adobe Photoshop and Adobe Illustrator CS3 were utilized solely for image assembly, ensuring that no adjustments were made to the fluorescence intensity. To ensure randomness and objectivity in our image selection, *esg*-GFP positive cells were selected randomly from the R4 region of the Drosophila midgut. Regions of Interest (ROIs) were also randomly chosen, with an area of $1.5 \times 10^4\,mM^2$ from the R4 region. The quantification of midgut cells was conducted using a Leica DM6-B microscope. The *esg*-GFP positive cell were randomly selected from R4 region of Drosophila midgut. ROIs were randomly selected of area $1.5 \times 10^4\,mM^2$ from R4 region of Drosophila midgut. For the analysis of immunofluorescence imaging results from the z stacks acquired with confocal microscopy, we employed ImageJ software, following the methodology previously described by Martin Fitzpatrick, which can be found at the following link: https://theolb.readthedocs.io/en/latest/imaging/measuring-cell-fluorescence-using-imagej.html.

Fluorescence intensity was quantified from confocal z-stack images captured utilizing a Leica TCS-SP8 microscope and processed with ImageJ. The process included opening the image, separating channels, setting the scale to pixels, configuring measurement options to include area and integrated density, selecting regions of interest

(ROIs), and calculating the adjusted integrated density using the following method: Integrated Density = (Integrated Density of ROI or cell - Integrated Density of background region)/ (Area of background region × Area of ROI or cell). This method provides a precise measurement of fluorescence within ROIs. For further inquiries or assistance, please contact us.

## Quantitative analysis of Western blot band intensity

Band grayscale values in Western blot images are quantified using ImageJ. Key steps include opening the image, inverting if needed, setting the scale to pixels, selecting the band area, and calculating the mean gray value. The formula for the band's grayscale value is the product of the mean gray value and area of the band minus the same for the background. This method provides a straightforward way to assess protein expression, factoring in background noise.

## RNA sequencing data analysis

We obtained RNA-seq data for T cells treated with CoA from the GEO repository (GSE187456) [37]. We selected CoA-treated Tc0 cells and untreated controls (GSM5668528, GSM5668529, GSM5668530, GSM5668531, GSM5668532, GSM5668533) for further analysis. GSEA was performed using the cloud-based GSEA version 19.0.25 (GenePattern cloud) with default parameters to investigate changes in key pathways for both CoA-treated and untreated Tc0 cells. GSEA was performed to investigate the changes in the pathways of interest for Tc0 cells + CoA and Tc0 cells untreated group.

## Single-cell RNA sequencing (scRNA-seq) data analysis

The *fbl* expression analysis was conducted using a user-friendly data portal. The complete set of assets can be obtained by visiting the AFCA webpage at https://hongjielilab.org/afca/ [50]. This platform allows users to explore gene-specific changes during aging and provides interactive tools for data analysis. Researchers can freely access aging trajectories and download necessary analytical assets through the AFCA interface.

## Supporting information

**S1 Data.  Primary data underlying graphs in figures.**
(XLSX)

**S1 Fig.  Reduced CoA synthesis in young ISCs results in deterioration of ISC functionality.** (A) Transcriptomic profiling of single-cell gene expression in *dPANK/fbl* between young and aged ISC populations. (B) Relative mRNA expression of *dPANK/fbl* from ISCs with aging. 14d is used as control and each dot represents repeating times. (C) Effects of different genotypes and age-old on CoA levels in ISCs from 14d flies with *UAS-LUC* RNAi (control), 40d with *UAS-LUC* RNAi, 14d with *UAS-dPANK/fbl* RNAi and 40d with *UAS-dPANK/fbl* driven by *esg-GAL4*. Each dot represents repeating times. (D) Quantification of pH3+ cells count from 14d old flies with indicated treatment. Each dot represents one midgut. (E-H) Representative photos of *esg*-GFP (green) and Dl+ (red) staining from flies with indicated administration from 14d flies. UAS-LUC RNAi is used as a control. (I) Quantification of the number percentage of *esg*+ Pros+ cells to total cells. Each dot represents one ROI from one midgut. *UAS-LUC* RNAi is used as a control. (J-M) Analysis of the ratio of Dl+ cells to total cells, number of ph3+ cells, ratio of *esg*-GFP+ cells to total cells, and ratio of NRE+ cells to *esg*+ cells per ROI in midguts during intestinal regeneration. *UAS-LUC* RNAi as a control. (N) Percentage survival rates of female flies using Canton-S as a wild-type under BLM induced chronic damage, with CoA supplementation and without it starting at 1–2-day old are presented. The numbers of quantified Drosophila: 100 for each group. Three independent experiments were

conducted. The findings are based on three separate experimental trials. Each dot represents one replicate times. *UAS-LUC* RNAi used as a control. Scare bar: 10µm. Error bars indicate SDs. ROI size: $1.5 \times 10^4$ µm². Asterisks denote levels of significance: *$p < 0.05$, **$p < 0.01$, ***$p < 0.001$, ****$p < 0.0001$, and ns (not statistically significant) signifies $p > 0.05$. One-way ANOVA with Tukey's multiple comparison test was used. Survival curves were analyzed using the log-rank (Mantel-Cox) test (N).
(TIF)

**S2 Fig. dPANK/fbl overexpression promotes ISC differentiation.** (A-B) Quantification of percentage of *esg*-GFP positive cells, DI positive cells to total cells, and number of pH3 positive cells per midgut with two independent *dPANK/fbl* RNAi lines per ROI. Each dot represents one midgut. (C) Representative photos with *esg*-GFP, NRE-lacZ and Pros staining from midguts of 14d with *UAS-LUC* (control), 40d flies with *UAS-LUC* flies and 40d flies with UAS-*dPANK/fbl* with indicated administration. (D-G) The count of pH3 + cells percentage of esg-GFP, DI, NRE-lacZ positive cells and *esg*+ pros+ cells to total cells per ROI is in S2C Fig. Each dot represents one midgut. Scare bar: 10µm. Error bars indicate SDs. ROI size: $1.5 \times 10^4$ µm². Asterisks denote levels of significance: *$p < 0.05$, **$p < 0.01$, ***$p < 0.001$, ****$p < 0.0001$, and ns (not statistically significant) signifies $p > 0.05$. One-way ANOVA with Tukey's multiple comparison test was used.
(TIF)

**S3 Fig. CoA counteracts the age-associated deterioration of intestinal functions and prolongs the life expectancy of *Drosophila*.** (A-B) Percentage survival rates of female flies (A) and male flies (B) and using Canton-S as a wild-type, with Pan supplementation and without it starting at 1–2-day old are presented. The numbers of quantified *Drosophila*: 100 for each group. Three independent experiments were conducted. The findings are based on three separate experimental trials. Each dot represents one replicate times. Error bars indicate SDs. Survival curves were analyzed using the log-rank (Mantel-Cox) test.
(TIF)

**S4 Fig. dPANK/fbl Function Cell Autonomously to Regulate Intestinal Stem Cell Differentiation in Drosophila.** (A) Typical images from midguts stained for *esg*-GFP (green) and DI (white) from 14d flies with *UAS-LUC* RNAi (control), *pros-GAL4* driven *UAS-dPANK/fbl* RNAi and *mex-GAL4* driven *UAS-dPANK/fbl* RNAi with the specified treatments indicated. (D, E) Measurement of the ratio of *esg*-GFP, DI, and number of pH3 counts to total cells in the S4A-C Fig () per ROI. Each dot represents one midgut. (F, G) Representative immunofluorescence photos (F) of midguts with Dlg staining and quantification Dlg expression (G) with indicated manipulation in experiments (K) and each dot represent one ROI from R4 midguts. (H) Quantification of intestinal acid-base equilibrium in *Drosophila* with indicated specific GAL4. *UAS-LUC* RNAi line was used as a control. (I) Quantification of the eating percent of *dPANK/fbl* flies driven by specific GAL4. Each dot represents one replicate times. (J) Quantification of deposits number from 14d flies with *UAS-LUC* RNAi (control), *UAS-dPANK/fbl* RNAi driven with specific GAL4. Each dot represents one quantified field from 20 Drosophila. (K) Quantification of percentage of "Smurf" of indicated treatment. (L) Assessment of dietary consumption in Drosophila through the CAFE assay from 14d flies with 14d flies with *UAS-LUC* RNAi (control), *UAS-dPANK/fbl* RNAi driven with specific GAL4 of indicated treatment. Error bars show the SD of three independent experiments. Each dot represents one replicate times. (M) Percentage survival rates of female flies and male flies and using Canton-S as a wild-type, with CoA supplementation and without it starting at 1–2-day old are presented. The numbers of quantified Drosophila: 100 for each group. Three independent experiments were conducted. The findings are based on three separate experimental trials. Each dot represents one replicate times. Scare bar: 10µm. Error bars indicate SDs. ROI size: $1.5 \times 10^4$ µm². Asterisks denote levels of significance: *$p < 0.05$, **$p < 0.01$, ***$p < 0.001$, ****$p < 0.0001$, and ns (not statistically significant) signifies $p > 0.05$. One-way ANOVA with

Tukey's multiple comparison test was used. Survival curves were analyzed using the log-rank (Mantel-Cox) test (K, L).
(TIF)

**S5 Fig. CoA promotes ISC-to-EC differentiation by inhibiting iron accumulation in ISCs via a ferroptosis-independent manner.** (A) Performing Gene Set Enrichment Analysis (GSEA) to compare the iron uptake and transport route between T lymphocytes subjected to CoA intervention and those that are non-treated. (B) Representative photos from midguts with FeRhoNox-1 staining (iron indicator) from 14d flies with *UAS-LUC* RNAi (control), and *UAS-dPANK/fbl* RNAi and *UAS-dPANK/fbl* RNAi with CoA treatment. (C) Quantification of fluorescence intensity of FeRhoNox-1 per ISCs from flies with indicated treatment. Each dot represents mean fluorescence intensity randomly chosen from 10 *esg*-GFP+ cells per ROI from 20 midguts. (D) Relative mRNA expression of ferroptosis associated genes in midguts of *dPANK/fbl* RNAi flies compared with controls (*UAS-LUC* RNAi) driven by *esg-GAL4*. Expression of control normalized to 1. n = 3 replicates. (E, F) Lipid peroxidation by BODIPY 581/591 C11 staining and quantification of C11 oxidation ratio of midguts. *UAS-LUC* RNAi was used as a control. (G, H) Immunostaining and quantification of 4-HNE (red) of midguts in *UAS-LUC* RNAi and *UAS-dPANK/fbl* RNAi flies Scare bar: 10µm. Error bars indicate SDs. ROI size: $1.5 \times 10^4$ µm$^2$. Asterisks denote levels of significance: *$p < 0.05$, **$p < 0.01$, ***$p < 0.001$, ****$p < 0.0001$, and ns (not statistically significant) signifies $p > 0.05$. Statistical analyses were conducted using Student's t-tests (H), One-way ANOVA with Tukey's multiple comparison test was used in other results.
(TIF)

**S6 Fig. CoA administration prevents ISC dysfunction through antioxidative ability and protects ISCs from stress-induced DNA damage.** (A) Representative photos of midguts with DHE staining from 14d flies with indicated administration. *UAS-LUC* RNAi was used as a control. (B) Quantification of DHE fluorescence intensity per ISCs from flies with indicated administration, (C) Representative photos of midguts with DHE staining from flies with indicated administration. (D) Quantification of DHE fluorescence intensity per ISCs from flies with indicated administration, 14d flies is used as control. Each dot represents mean fluorescence intensity randomly chosen from 10 esg-GFP+ cells per ROI and total of 21 midguts. Scare bar: 10µm. Error bars indicate SDs. ROI size: $1.5 \times 10^4$ µm$^2$. Asterisks denote levels of significance: *$p < 0.05$, **$p < 0.01$, ***$p < 0.001$, ****$p < 0.0001$, and ns (not statistically significant) signifies $p > 0.05$. One-way ANOVA with Tukey's multiple comparison test was used.
(TIF)

## Acknowledgments

We thank BDSC, VDRC, and Tsinghua Fly Center for fly strains and DSHB for antibodies.

## Author contributions

**Conceptualization:** Zhiming Liu, Gang Du, Haiyang Chen.

**Funding acquisition:** Yi Chen, Haiyang Chen.

**Investigation:** Zhiming Liu, Gang Du.

**Methodology:** Zhiming Liu, Gang Du.

**Resources:** Yi Chen, Haiyang Chen.

**Supervision:** Yi Chen, Haiyang Chen.

**Validation:** Zhiming Liu.

**Visualization:** Gang Du.

**Writing – original draft:** Zhiming Liu, Yi Chen, Haiyang Chen.

**Writing – review & editing:** Zhiming Liu, Gang Du, Yi Chen, Haiyang Chen.

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
