## [Decision Letter · Decision Letter 0]

22 Dec 2024

PGENETICS-D-24-01387

Age-associated Decline of Coenzyme A leads to Intestinal stem cells dysfunction via disturbing iron homeostasis

PLOS Genetics

Dear Dr. Chen,

Thank you for submitting your manuscript to PLOS Genetics. After careful consideration, we feel that it has merit but does not fully meet PLOS Genetics's publication criteria as it currently stands. Therefore, we invite you to submit a revised version of the manuscript that addresses the points raised during the review process.

Please submit your revised manuscript within 60 days Feb 20 2025 11:59PM. If you will need more time than this to complete your revisions, please reply to this message or contact the journal office at plosgenetics@plos.org. Please include the following items when submitting your revised manuscript:

We look forward to receiving your revised manuscript.

Kind regards,

Hongyan Wang, Ph.D.

Academic Editor

PLOS Genetics

Fengwei Yu

Section Editor

PLOS Genetics

Aimée Dudley

Editor-in-Chief

PLOS Genetics

Anne Goriely

Editor-in-Chief

PLOS Genetics

**Journal Requirements:**

At this stage, the following Authors/Authors require contributions: Haiyang Chen. Please ensure that the full contributions of each author are acknowledged in the "Add/Edit/Remove Authors" section of our submission form.

The list of CRediT author contributions may be found here: https://journals.plos.org/plosgenetics/s/authorship#loc-author-contributions

https://journals.plos.org/plosgenetics/s/submission-guidelines#loc-parts-of-a-submission

5) We notice that your supplementary Figures are included in the manuscript file. Please remove them and upload them with the file type 'Supporting Information'. Please ensure that each Supporting Information file has a legend listed in the manuscript after the references list.

Potential Copyright Issues:

i) Please confirm (a) that you are the photographer of 4A, and 4F, or (b) provide written permission from the photographer to publish the photo(s) under our CC BY 4.0 license.

ii) Figures 1A, and 6L. Please confirm whether you drew the images / clip-art within the figure panels by hand. If you did not draw the images, please provide (a) a link to the source of the images or icons and their license / terms of use; or (b) written permission from the copyright holder to publish the images or icons under our CC BY 4.0 license. Alternatively, you may replace the images with open source alternatives. See these open source resources you may use to replace images / clip-art:

7) Please amend your detailed Financial Disclosure statement. This is published with the article. It must therefore be completed in full sentences and contain the exact wording you wish to be published.

**Reviewers' comments:**

Reviewer's Responses to Questions

Reviewer #1: In this manuscript, the authors demonstrate that reduced activity of the pantothenate/coenzyme A (CoA) pathway in aged Drosophila ISCs provides a novel and specific insight into the metabolic changes associated with aging. However, the study lacks depth in explaining how pantothenate/CoA supplementation mechanistically restructures ISC metabolism or reduces iron accumulation. Overall, this is an interesting paper and I have a few suggestions to further improve this manuscript:

1. The manuscript mentions a "screening strategy" but does not elaborate on its nature or methodology. Providing a brief description—such as whether the screening was genetic, metabolic, or otherwise—would help readers better understand the study’s approach and scope.

2. The authors show that reduced CoA synthesis in young ISCs results in compromised ISC functionality. However, it remains unclear whether the regulatory effect of CoA on ISC proliferation is cell-autonomous or non-cell-autonomous. For example, would disrupting CoA synthesis in enterocytes (ECs) or enteroendocrine cells (EEs) also influence ISC proliferation?

3. Similarly, it would be useful to determine if perturbing CoA synthesis in ECs or EEs affects age-related declines in intestinal functions and overall lifespan, as observed with alterations in ISCs.

4. The data in Figure 5 show that loss of dPANK/fbl leads to a substantial increase in iron levels in both ISCs and ECs. The authors should discuss possible mechanisms driving this iron accumulation—such as altered iron import, storage, or export—and explain how these changes might impact ISC functionality and overall intestinal homeostasis.

5. Elevated iron levels can facilitate the Fenton reaction, thereby increasing ROS production and potentially leading to lipid peroxidation and ferroptosis. The authors should clarify whether indicators of ferroptosis—such as lipid peroxidation-induced membrane damage or changes in ferroptosis-related gene expression—are detectable in dPANK/fbl-depleted gut epithelium. Such information would provide insights into the downstream consequences of iron dysregulation in this model.

Reviewer #2: In this paper, Liu et al. identified an interesting phenotype: endogenous coenzyme A (CoA) levels in the adult fly midgut decrease with age. They proposed that this decline in CoA correlates with intestinal stem cell (ISC) hyperproliferation, a phenotype commonly observed in aging flies. Supporting this hypothesis, they demonstrated that feeding older flies with CoA could suppress ISC hyperproliferation associated with aging. Furthermore, the authors showed that knocking down dPANK/fbl, a critical gene in the CoA synthesis pathway, promotes ISC proliferation, whereas CoA supplementation counteracts this over-proliferation caused by dPANK/fbl depletion. Overall, this section of the paper is compelling, and the genetic experiments are well-designed and logically executed. Identifying Coenzyme A (CoA) as a therapeutic target to counteract ISC aging is innovative and could have broad implications for anti-aging therapies. However, the study has several significant flaws that undermine its publishability. For example, the authors concluded that CoA supplementation promotes ISC differentiation and rescues the differentiation defects caused by dPANK/fbl depletion. Unfortunately, the data presented in Figures 2 and 3K-P contradict this conclusion, as discussed in the following comments. Additionally, the conclusions drawn from Figures 5A-B, 5E-F, and 6A-B are problematic. Unfortunately, because of these shortcomings, we cannot recommend this paper for publication in PLOS Genetics in its current form. However the topic is significant, and with substantial revisions (including new data), to address these issues the paper could be a valuable addition to the literature on stem cell functions and aging. Detailed comments are provided below.

Major points:

1. As mentioned, the data shown in Figures 2 and 3K–P contradict the conclusion that CoA supplementation promotes ISC differentiation. If CoA administration indeed promotes ISC differentiation, there should be an increase in esg+mex+(Figures 2B-E), esg+pros+ (Figures 2F-I), and NRE-lacZ+ (Figures 2F-H, J) cells in the “40d+CoA” group compared to the “40d” group. Referencing typical pro-differentiation gene phenotypes (e.g., Notch overexpression) from previous publications might be helpful. However, the observed phenotype suggests the opposite, potentially indicating that CoA supplementation impairs ISC self-renewal (proliferation) and regenerative capacity. A similar misinterpretation applies to Figures 3K–P, where the data suggest that CoA blocks dPANK/fbl RNAi-induced pro-mitotic effects. Additional data are needed to rule out this alternative interpretation.

2. The immunofluorescence data shown in Fig. 5A-B and 5E-F, and Fig. 6A-B are sloppy, and the conclusions made on them are unreliable. How did the authors determine that the observed increase in RhoNox-1 in ISCs was not due to technical inconsistencies in staining across parallel groups? Notably, dPANK/fbl RNAi guts exhibit significantly higher overall RhoNox-1 levels. Additionally, if dPANK/fbl functions specifically in ISCs, why does dPANK/fbl knockdown increase RhoNox-1 both autonomously and non-autonomously? This requires clarification.

3. Following the above point #2, While using fluorescence intensity quantification to demonstrate significant differences among groups is a standard approach, it raises ethical concerns. Specifically, were the images selected arbitrarily? This concern is particularly relevant when phenotypes are subtle and quantification is necessary to distinguish them. The authors should provide details about the image selection process and declare how the images were chosen for quantification.

4. The data in Figures 1K-L suggest an intriguing possibility: that CoA restrains ISC proliferation (and gut aging) via a cell-autonomous mechanism or by preventing intestinal barrier disruption, which has been shown to cause ISC hyperproliferation in previous studies. This distinction needs further investigation.

5. The authors should examine whether dPANK/fbl overexpression increases CoA levels in the gut (Figure 3C) and whether dPANK/fbl overexpression prevents ISC proliferation in aging guts.

6. Fig. 4D, the authors should show the significance between “40d” and “40d+CoA” groups.

7. Fig. 4G-4J, using “eating percentage” to evaluate gut function seems speculative. The authors should provide citations to support this as a reliable marker.

8. There is only one example of lifespan extension (Fig 4K). Since lifespan extension is a major point of the paper, more lifespan data should be included. Examples from both sexes, with different levels of CoA and Pantothenate supplementation could be provided. This lifespan data should be extensive and conclusive. Please also address the relatively modest effects of CoA on lifespan in Figure 4.

9. The figure legends need to be revised to provide clear, concise, and complete descriptions of the data. Crucially, any experimental details missing in the Results section should be included in the figure legends to ensure that the figures are independently interpretable. For example, details about specific controls, experimental conditions, or quantification methods should be described in the legends if omitted elsewhere in the manuscript.

10. Please elaborate on the mechanistic link between CoA’s antioxidative properties and its ability to modulate ISC proliferation and differentiation.

Minor points:

1. In the introduction, it would be helpful to review previous studies examining whether defects in the CoA biosynthesis pathway correlate with colorectal cancer progression.

2. Also in the introduction, the authors mentioned that “using a screening strategy, we found that the pantothenate/coenzyme A (CoA) pathway is abundant in young ISCs and identified CoA as an inhibitor of age-related ISC over-proliferation.” Please provide a little bit more details about this screen in the Method section.

3. Please define ROIs and the number of regions analyzed for quantification, particularly in figures that rely on spatial specificity (e.g., Figures 1J, 3E, 5D, 6D).

4. Please reduce nonspecific background staining in image-based quantifications and explain the steps to ensure reliable measurements.

5. Please provide detailed information on the ages of flies used in all experiments, especially in Figures 3C and 3D.

6. The abbreviation “FAC” should be defined where it is first introduced (page 9).

Reviewer #3: Drosophila intestinal stem cells (ISCs) represent a powerful model to elucidate the mechanisms controlling and executing regeneration associated with aging fueled by somatic stem cells. In their manuscript ‘Age-associated Decline of Coenzyme A leads to Intestinal stem cells dysfunction via disturbing iron homeostasis’, The work described by Liu and colleagues identified and report Coenzyme A(CoA) as a key driver in regenerative and aging associated dysfunction of the adult Drosophila midgut with a screening approach. Furthermore, they show that the CoA regulated aged ISC dysfunction through regulating ISC iron availability. It represents a significant gain in the understanding of somatic stem cell-driven tissue repair and regeneration in the fly gut. The described metabolic mechanism may operate similarly in other examples of stem cell function and, more generally, in diverse examples of aging associated tissue dysfunction. The manuscript is well written and data is comprehensively presented. The data is based on state-of-the-art methods and provides new important insights in the mechanisms of stem cell differentiation and tissue regeneration associated with aging. General interest is high. Publication as a PLOS Genetics paper is appropriate.

Although the authors present data on their findings, I have a few suggestions and concerns that I think should be approached experimentally and discussed in more detail in a revised version of this manuscript.

Minor comment:

1、Could the authors provide speculation on how the CoA regulated iron homeostasis during the fbl is depleted, allowing the ROS to be accumulated?

2、The sequence of the figures is not intuitive and could be confusing for readers. Please consider rearranging the order of the figures for better clarity. For instance, figures 2F-2H should be moved to the left. Additionally, it is recommended to place statistical graphs on the right side.

3、Reference to previous work on has emerged as a potential strategy for enhancing stem cell functionality (4). The reference cited is not of an earlier publication; please search for and cite more earlier literature account of metabolism in stem cells.

4、In Figure 3, to test whether fbl RNAi knockdown in ISCs of young fruit flies would result in a stem cell aging phenotype, the authors employed only one fbl RNAi line to examine the phenotype following knockdown in stem cells. Is there any possibility that off-target effects from the RNAi could be responsible for this? Some of the central conclusions made in the study do require additional data to be substantiated.

5、This figure has hardly any esg-GFP+ cells signals in the 14 day condition. While the number esg-GFP+ cells of flies with fbl knockdown or 40 day is robust, the absence of esg-GFP+ looks weird. Please comment, and add a description in the text

6、Figure 5A: why does the ferric iron signal have a dotted pattern of young flies with fbl knockdown in Figure 5, and its signal is not observed in young flies? This is also partially observed in the old flies (Fig 5E).

**Have all data underlying the figures and results presented in the manuscript been provided?**

Reviewer #1: Yes

Reviewer #2: Yes

Reviewer #3: Yes

PLOS authors have the option to publish the peer review history of their article (what does this mean? ). If published, this will include your full peer review and any attached files.

**Do you want your identity to be public for this peer review?** For information about this choice, including consent withdrawal, please see our Privacy Policy .

Reviewer #1: No

Reviewer #2: **Yes: ** Bruce Edgar

Reviewer #3: **Yes: ** Zhouhua Li

**Figure resubmission:**
---

## [Decision Letter · Decision Letter 1]

29 Apr 2025

Dear Dr Chen,

We are pleased to inform you that your manuscript entitled "Age-associated Decline of Coenzyme A leads to Intestinal stem cells dysfunction via disturbing iron homeostasis" has been editorially accepted for publication in PLOS Genetics. Congratulations!

Yours sincerely,

Hongyan Wang, Ph.D.

Academic Editor

PLOS Genetics

Fengwei Yu

Section Editor

PLOS Genetics

Aimée Dudley

Editor-in-Chief

PLOS Genetics

Anne Goriely

Editor-in-Chief

PLOS Genetics

Comments from the reviewers (if applicable):

Reviewer's Responses to Questions

**Comments to the Authors:**

Reviewer #1: The author has addressed all my comments.

Reviewer #3: The authors had addressed all the concerns I raised.

**Have all data underlying the figures and results presented in the manuscript been provided?**

Reviewer #1: Yes

Reviewer #3: Yes

PLOS authors have the option to publish the peer review history of their article (what does this mean? ). If published, this will include your full peer review and any attached files.

**Do you want your identity to be public for this peer review?** For information about this choice, including consent withdrawal, please see our Privacy Policy .

Reviewer #1: No

Reviewer #3: **Yes: ** Zhouhua Li

**Data Deposition**

http://datadryad.org/submit?journalID=pgenetics&manu=PGENETICS-D-24-01387R1

**Press Queries**

---

## [Editor Report · Acceptance letter]

PGENETICS-D-24-01387R1

Age-associated Decline of Coenzyme A leads to Intestinal stem cells dysfunction via disturbing iron homeostasis

Dear Dr Chen,

We are pleased to inform you that your manuscript entitled "Age-associated Decline of Coenzyme A leads to Intestinal stem cells dysfunction via disturbing iron homeostasis" has been formally accepted for publication in PLOS Genetics! Your manuscript is now with our production department and you will be notified of the publication date in due course.

With kind regards,

Anita Estes

PLOS Genetics

On behalf of:
